# Two receptor tyrosine phosphatases dictate the depth of axonal stabilizing layer in the visual system

Satoko Hakeda-Suzuki*, Hiroki Takechi, Hinata Kawamura, Takashi Suzuki*

School of Life Science and Technology, Tokyo Institute of Technology, Yokohama, Japan

**Abstract** Formation of a functional neuronal network requires not only precise target recognition, but also stabilization of axonal contacts within their appropriate synaptic layers. Little is known about the molecular mechanisms underlying the stabilization of axonal connections after reaching their specifically targeted layers. Here, we show that two receptor protein tyrosine phosphatases (RPTPs), LAR and Ptp69D, act redundantly in photoreceptor afferents to stabilize axonal connections to the specific layers of the *Drosophila* visual system. Surprisingly, by combining loss-of-function and genetic rescue experiments, we found that the depth of the final layer of stable termination relied primarily on the cumulative amount of LAR and Ptp69D cytoplasmic activity, while specific features of their ectodomains contribute to the choice between two synaptic layers, M3 and M6, in the medulla. These data demonstrate how the combination of overlapping downstream but diversified upstream properties of two RPTPs can shape layer-specific wiring.
DOI: https://doi.org/10.7554/eLife.31812.001

## Introduction

Receptor protein tyrosine phosphatases (RPTPs) play a pivotal role in the development and function of the nervous system by regulating the activity, localization, and binding of various molecules. The extracellular components of RPTPs are diverse and grouped into eight subtypes (*Johnson and Van Vactor, 2003*). Type IIA RPTPs contain extracellular immunoglobulin-like (Ig) domains, fibronectin type III domains (FNIII), and two intracellular phosphatase domains. Three type IIA RPTPs in vertebrates, LAR, PTP-δ, and PTP-σ are known to be involved in nerve growth, guidance, regeneration, and synaptogenesis (*Ensslen-Craig and Brady-Kalnay, 2004*; *Stoker, 2015*).

Of six RPTPs found in *Drosophila*, two phosphatases, LAR and Ptp69D have a similar structure and are classified as Type IIA RPTPs. These Drosophila phosphatases have been extensively studied in the nervous system. It has been shown that several RPTPs, including LAR and Ptp69D, have some overlapping functions in the *Drosophila* nervous system (*Desai et al., 1997*; *Jeon et al., 2008*; *Sun et al., 2000*; *Sun et al., 2001*). For example, Ptp10D and Ptp69D exhibit overlapping function in CNS axon guidance (*Sun et al., 2000*). Most ISNb motor axons have normal projections in Ptp69D or Ptp99A single mutants, but double mutants have severe axonal defects (*Desai et al., 1996*). Similarly, both LAR and Ptp69D play an important role during axon guidance and target selection in the embryonic CNS and PNS (*Desai et al., 1997*; *Krueger et al., 1996*; *Sun et al., 2000*) and display considerable overlapping function (*Desai et al., 1997*).

The *Drosophila* visual system has a layered structure, as is commonly observed both in vertebrates and invertebrates. Each layer is specifically innervated by processes from specific sets of afferent neurons. Moreover, formation of a functional neuronal network requires not only precise target neuron recognition, but also stabilization of the axonal contacts within their appropriate synaptic layer during development. Owing to its simple and stereotyped structure with innervations from

*For correspondence:
hakeda@bio.titech.ac.jp (SH-S);
suzukit@gmail.com (TS)

**Competing interests:** The authors declare that no competing interests exist.

elaborate arrays of distinct photoreceptor axons, the *Drosophila* visual system is a widely used model for exploring the molecular mechanisms underlying the wiring of layer-specific connections. In particular, the medulla, which is the second ganglion of *Drosophila* optic lobe, has characteristic laminar structure and is divided into ten layers, M1 to M10. The axons from photoreceptors R7 and R8 terminate at the layer M6 and M3, respectively.

RPTPs have been shown to be important for layer-specific targeting. LAR and Ptp69D are required for R7 axons to create proper connections in medulla layer M6 (*Newsome et al., 2000a*). In *LAR* mutants, R7 axons initially target to the correct layer at early pupal stages, but they later retract to the R8 axon temporary layer, M3 (*Clandinin et al., 2001*; *Maurel-Zaffran et al., 2001*). The degree of functional redundancy between LAR and Ptp69D in selection of the final targeting layer (M6 versus M3) has also been assessed previously. Rescue experiments revealed that LAR can substitute for Ptp69D, but not vice versa (*Maurel-Zaffran et al., 2001*). In addition, it has been suggested that LAR phosphatase activity is not required for R7 targeting depending on Ptp69D phosphatase activity (*Hofmeyer and Treisman, 2009*). These findings indicate that LAR and Ptp69D have both highly similar and divergent functions, but their distinct versus overlapping effects on downstream signaling pathways governing layer specification of R7 axons remain unclear.

In this study, we show that the R7 axons lacking both LAR and Ptp69D display a striking phenotype: termination in the lamina without innervation to the medulla. Thus, building on previous studies describing roles of LAR and Ptp69D in layer M3 versus M6 determination within the medulla, we show here that they have a crucial function for multi-layer specification in wider range from lamina to the medulla. Moreover, characterization of R7 axon extension over pupal development revealed that the double mutant R7 axons extend normally at the beginning of the pupal stage, but gradually retract from their correct temporary layer. Genetic manipulations leading to graded expression of LAR and Ptp69D indicated that R7 axons retract either to the lamina, surface of the medulla (referred to as M0 as described in [*Akin and Zipursky, 2016*]), M3, or M6 (as in wild type), according to the total expression level of each gene. Combining loss-of-function and genetic rescue experiments, we demonstrated that the depth of the final axon termination layer is controlled by (1) the cumulative amount of the cytoplasmic signaling activity of LAR and Ptp69D, and (2) distinct layer-specific expression patterns of ligands selective to the ectodomains of either LAR or Ptp69D. These data demonstrate how the convergence of common downstream signals generated by receptors with redundant downstream targets but different upstream ligands can shape layer-specific wiring in the brain.

## Results

### LAR and Ptp69D have redundant functions in layer-specific targeting of R7 axons

To determine if LAR and Ptp69D have overlapping functions in layer-specific targeting and stabilization of R7 axons, we conceived a strategy to knock them out simultaneously in photoreceptors using the FLICK system (*Hakeda-Suzuki et al., 2011*) combined with null alleles of *LAR* and *Ptp69D*. We first applied the FLICK system to generate *LAR* and *Ptp69D* single knockout mutants (see Methods) and confirmed the phenotype of each (*Figure 1*). As reported previously, 90% of the R7 axons from the *LAR* single mutant and 20% of the R7 axons from the *Ptp69D* single mutant photoreceptors stabilized inappropriately at M3 (*Figure 1A–C and H*) (*Hofmeyer and Treisman, 2009*; *Maurel-Zaffran et al., 2001*; *Newsome et al., 2000a*). The lower penetrance of the *Ptp69D* phenotype here is probably due to the low efficiency of the FLICK system. Next, we generated double-knockout FLICK mutants. Surprisingly, more than 80% of the R7 axons of *LAR, Ptp69D* double mutants failed to innervate the medulla and terminated inside the lamina (*Figure 1D–H*, See Materials and methods for quantification details). To exclude the possibility of R7 cell death, we checked agarose sections of double mutant retina for the existence of R7 cell bodies, and whether the axons were extending from those cell bodies (*Figure 1E and F*). These results indicate that LAR and Ptp69D share a common function in R7 axons to drive innervation to the medulla.

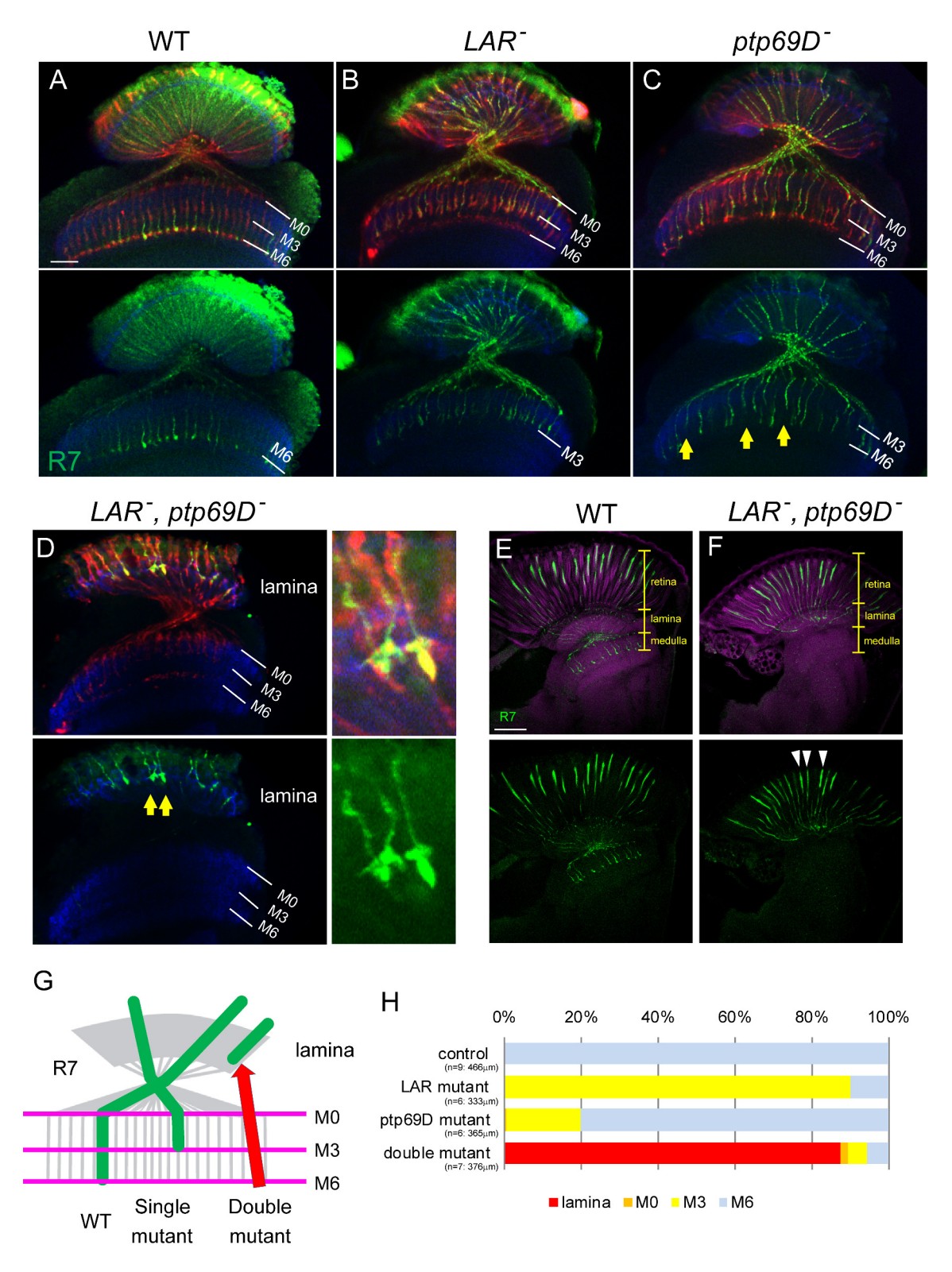

**Figure 1.** Defects of R7 axons in the *Lar, Ptp69D* double mutant. (**A–D**) Horizontal image of the adult medulla; WT (**A**), *LAR* mutant (**B**), *Ptp69D* mutant (**C**) and *LAR, Ptp69D* double mutant (**D**). Photoreceptor axons were labeled with mAb24B10 (red), R7 photoreceptor axons with Rh4-mCD8GFP (green), and medulla layers with an antibody to N-Cadherin (blue). Medulla layers are indicated with white lines. In WT, all R7 axons terminated at layer M6 (**A**). 90% of R7 axons in *LAR* mutant (**B**), and 20% of R7 axons in *Ptp69D* mutant (**C**, arrows) terminated improperly in layer M3. In *Lar, Ptp69D* double
*Figure 1 continued on next page*

*Figure 1 continued*

mutants, R7 axons did not innervate the medulla and instead terminated inside the lamina (D: arrows). The magnified images of R7 terminals are shown on the right side. (E, F) Horizontal agar section of WT (E) and *Lar, Ptp69D* double mutant (F), labeled with Rh4-mCD8GFP (green) and N-Cadherin (magenta). R7 axons extending from their cell bodies (arrowheads) can be observed in the retina of the double mutant (F). (G) A schematic drawing of the phenotype of R7 axons in single mutants and double mutant of *LAR* and *Ptp69D*. (H) Quantification of the R7 axons terminating in each layer for each genotype. The number of axons terminating inside the lamina was estimated. See Methods for quantification details. Scale bars: (A) 20 μm (E) 50 μm.

DOI: https://doi.org/10.7554/eLife.31812.002
The following source data is available for figure 1:

**Source data 1.** Excel file compiling source data for the *Figure 1H*.
DOI: https://doi.org/10.7554/eLife.31812.003

## Either LAR or Ptp69D is required for creating stable connections in the R7 temporary layer

It has been reported that, in *LAR* mutants, R7 axons target the correct R7 temporary layer at the early pupal stage, but later retract to the R8 recipient layer (*Clandinin et al., 2001*, *Ting et al., 2005*). To assess the course that R7 axon terminals take to reach their final layer in double mutants, we observed R7 axons at several stages of pupal development (*Figure 2*). As the FLICK double mutation was often developmentally lethal, we were not able to obtain sufficient numbers of pupae with the correct genotype for these analyses. Therefore, we knocked down *LAR* and *Ptp69D* with eye specific RNAi driven by GMR-Gal4 in the flies with one copy of both genes removed. The R7 axons were marked by R7-specific Flippase (20C11FLP) (*Chen et al., 2014*) and UAS-FSF-mCD8GFP which was expressed under the GMR-Gal4 driver.

The adult phenotype of the R7 axons in these flies was similar to the FLICK double mutants (*Figure 2G'*). As in the *LAR* mutants, most of the R7 axon terminals reached their proper layer up to 24 hr after puparium formation (24hrAPF) (*Figure 2A–B'*). Around 36hrAPF, the shape of the growth cones was thinner and several axons had started to retract (*Figure 2C and C'*). At 42hrAPF, we found most of the axon terminals were collapsed and detached from the R7 temporary layer (*Figure 2D and D'*). At 48hrAPF hardly any axons remained inside the medulla (*Figure 2E and E'*) and this persisted through adulthood (*Figure 2F–G'*). These observations suggest that LAR and Ptp69D have a common role to promote attachment and stabilization of axons in the R7 temporary layer during the development of the medulla.

We then assessed requirements for LAR and Ptp69D at specific time points in development. To test the temporally regulated roles of LAR and Ptp69D, we modulated the levels of LAR and Ptp69D expression at several pupal developmental stages using temperature sensitive Gal80 (Gal80$^{ts}$). For this, we drove *LAR* and *Ptp69D* RNAi simultaneously in photoreceptors by GMR-Gal4, which resulted in the termination of R7 axons at the surface of the medulla, M0 (*Figure 2—figure supplement 1*). When the LAR and Ptp69D expression level was recovered at 25%APF, R7 axons stabilized in the proper layer M6. However, when we restored LAR and Ptp69D expression levels at 50%APF and 75%APF, R7 axons, especially in posterior region, mostly terminated in the M0 (*Figure 2—figure supplement 1*). In the converse experiment, decreasing the expression level at 12%APF or 25%APF caused some R7 axons to retract, but this did not occur upon knockdown at 50%APF (*Figure 2—figure supplement 1*). These results indicate that LAR and Ptp69D are required between 25%APF and 50%APF to promote axonal attachment and stabilization, which coincides with the period that R7 axons start to retract in the double mutant.

## The expression level of LAR and Ptp69D determines the final layer of axon termination and stabilization

We found that some of the R7 axons in single *LAR* and *Ptp69D* mutants stably terminated in M3, whereas the axons of double mutants retracted to the lamina. These results led us to postulate that the total level of LAR and Ptp69D phosphatase activity might determine the final depth of R7 axons. To examine this hypothesis, the expression level of each gene was manipulated simultaneously. (*Figure 3*, *Figure 3—figure supplement 1*). First, to achieve the highest expression of each gene, we suppressed the expression of double RNAi by temperature sensitive Gal80$^{ts}$ at permissive

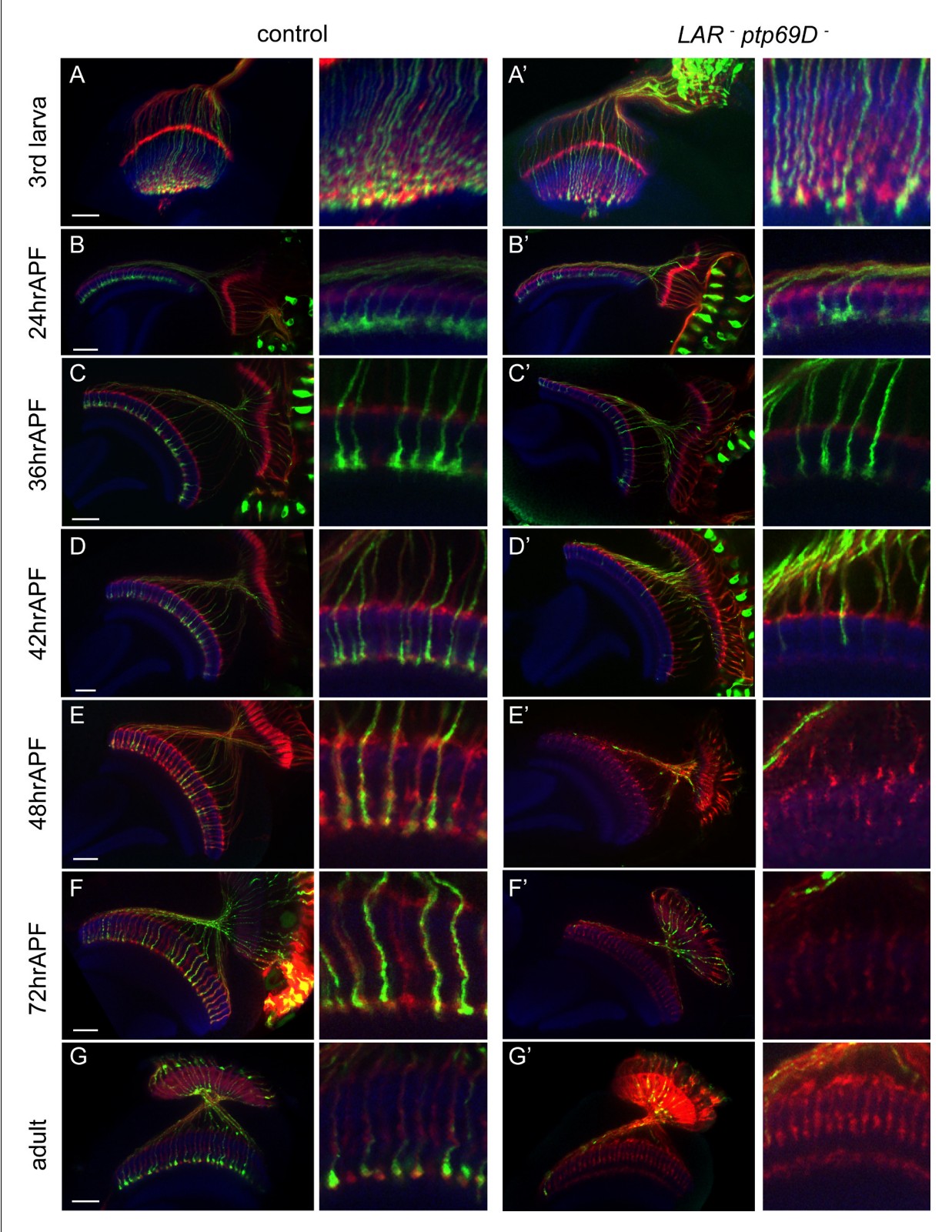

**Figure 2.** Retraction of R7 axons in the *LAR, Ptp69D* double mutant. (**A–G'**) The R7 axons of control (**A–G**) and *LAR, Ptp69D* double heterozygote with double RNAi (**A'–G'**) during the pupal stage were visualized with mCD8GFP (green) and counterstained with mAb24B10 (red) and N-Cadherin (blue). More than two samples were analyzed for each stage. Higher magnification images are shown on the right side of each panel. In late third instar larvae (**A, A'**) and pupae at 24hrAPF (**B, B'**), R7 axons of double mutant reached the R7 temporary layer correctly. At 36hrAPF (**C, C'**), R7 axon terminals started
*Figure 2 continued on next page*

*Figure 2 continued*

to get thinner. At 42hrAPF (**D, D′**), growth cones had collapsed and retraction of R7 axons was observed. At 48hrAPF (**E, E′**), almost all the R7 axons retracted out of medulla and remained in that location into adulthood (**F–G′**). Scale bars: 20 µm.

DOI: https://doi.org/10.7554/eLife.31812.004

The following figure supplement is available for figure 2:

**Figure supplement 1.** Temporal requirement of LAR and Ptp69D.

DOI: https://doi.org/10.7554/eLife.31812.005

temperature (23°C), which resulted in almost normal targeting phenotype. In contrast, to achieve the lowest expression, the heterozygous mutants of *LAR* and *Ptp69D* that also expressed RNAi for each gene without Gal80$^{ts}$ at 25°C were used, which displayed as strong a phenotype as homozygous mutants (***Figure 3E and F***). Intermediate RNAi expression levels were set by housing flies at a series of different temperatures (see Materials and methods for details). In this way, we were able to control the expression level of LAR and Ptp69D in a temperature controllable manner and thereby generated a series of mutants with graded expression levels of each gene (***Figure 3***, ***Figure 3—figure supplement 1***). At the lowest expression level, most of the R7 axons retracted to the lamina, and the number of R7 axons stabilizing appropriately in the medulla gradually increased as the expression level of LAR and Ptp69D was increased through temperature control (***Figure 3***). Two key conclusions can be deduced from this experiment. First, the depth of the final axon termination layer is controlled by the cumulative expression level of LAR and Ptp69D. Second, the final layer of axon termination and stabilization determined by LAR and Ptp69D levels seems to shift in a step-wise manner, M6 to M3 to M0 to the lamina layer in decreasing order of expression, thus indicating that there are discontinuous adhesive properties across the layers of the medulla. These results suggest that LAR and Ptp69D play both redundant and additive roles in stabilizing R7 axons in the medulla and determining the final layer position.

## LAR and Ptp69D cytoplasmic activity correlates with the depth of the layer of termination

We next asked what region of the LAR and Ptp69D proteins was responsible for determining the layer of termination: cytoplasmic phosphatase activity or ectodomains? To dissect the functional similarity and difference between the domains of LAR and Ptp69D, we tested requirements for their catalytic activity by selectively expressing modified or wild type *LAR* and/or *Ptp69D* transgenes in the photoreceptors of the double mutants (***Figure 4***, ***Figure 4—figure supplement 1***). LAR has two catalytic domains (D1 and D2) and a catalytically obligatory cysteine was mutated to serine in phosphatase-dead *LAR* transgenes (***Krueger et al., 2003***). As for Ptp69D, aspartate residues in the catalytic domain have been shown to be essential. When these residues were changed to alanine, it resulted in a mutant protein that acts as a substrate trap, which can bind but not efficiently hydrolyze phosphotyrosine (***Garrity et al., 1999***).

The projection defects observed in our double mutants were fully rescued by introducing unmodified full length LAR, with almost all the R7 axons attaching to their correct target, M6 (***Figure 4B and M***). This result is consistent with previous findings that R7 mistargeting in a *Ptp69D* single mutant can be rescued by full length LAR (***Maurel-Zaffran et al., 2001***). Next, we examined the ability of mutated forms of LAR to rescue the phenotype. Deletion of the extracellular LAR FnIII7-9 domains could not rescue the double mutant phenotype, indicating that FnIII domains are essential to stabilize R7 axons inside the medulla (***Figure 4F and M***). Not surprisingly, we also found that the existence of LAR FnIII7-9 was sufficient to rescue the double mutant phenotype (***Figure 4G, H and M***); its presence was previously reported to be sufficient for rescuing LAR single mutant (***Hofmeyer and Treisman, 2009***). We then tested a series of mutated cytoplasmic domain fragments, which in all cases retained the normal LAR ectodomain (***Krueger et al., 2003***). Specifically, we assessed transgenes with the second or both phosphatase domains deleted (***Figure 4C and M***), and phosphatase-dead mutations affecting one or both phosphatase domains (***Figure 4D, E and M***). R7 axons gradually shifted their final termination layer depending on the strength of the mutation in the cytoplasmic domain (***Figure 4M***): the stronger the mutation, the shallower the layer of termination. The double phosphatase-dead (2xCS) construct rescued 40% of the R7 axons projecting

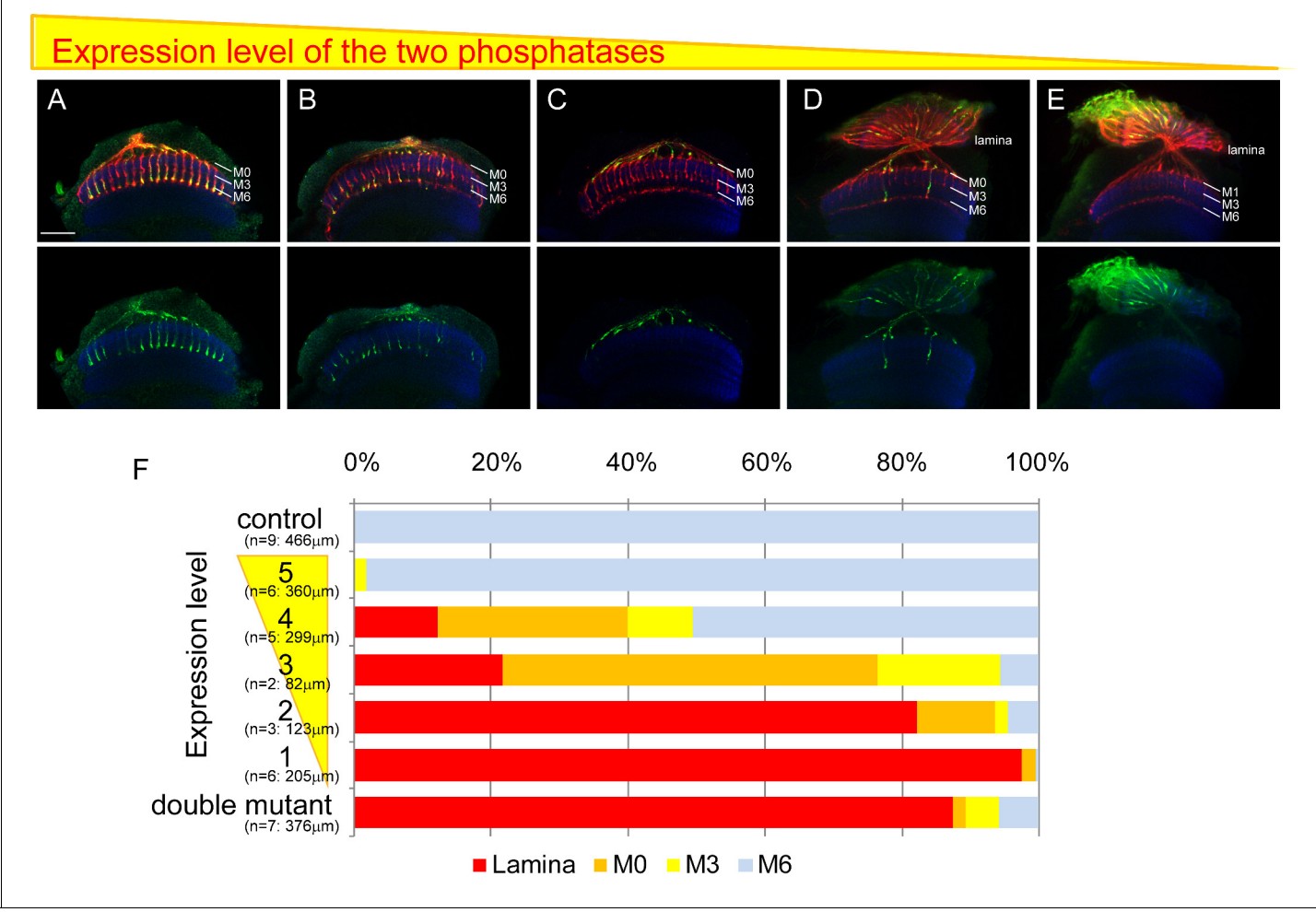

**Figure 3.** Final stabilizing layer determinations by LAR and Ptp69D. (A–E) Genetic manipulation using a temperature sensitive Gal80[ts] system driving RNAi expression was employed to manipulate the cumulative expression level of LAR and Ptp69D. (A–C) *LAR, Ptp69D* RNAi transcripts were expressed using the eye specific GMR-Gal4 line and RNAi expression level was controlled using Gal80[ts] (A: 23°C, B: 25°C, C: 27°C). (D, E) *LAR, Ptp69D* RNAi was simultaneously expressed using eye specific GMR-Gal4 in the *Lar⁻/+, Ptp69D⁻/+* background (D: 18°C E: 25°C). Photoreceptor axons are labeled with mAb24B10 (red), R7 photoreceptor axons with Rh4-mCD8GFP (green), and the medulla layers with an antibody to N-Cadherin (blue). The medulla layers are indicated with white lines. (F) Quantification of the R7 axons terminated in each layer. The depths of the R7 axons depicted in A-E were quantified and defined according to expression levels of 5 to 1 (A=5, B=4, C=3, D=2, E=1). The number of axons terminating inside the lamina was estimated as in *Figure 1*. Scale bar: 20 µm.

DOI: https://doi.org/10.7554/eLife.31812.006

The following source data and figure supplement are available for figure 3:

**Source data 1.** Excel file compiling source data for the *Figure 3F*.
DOI: https://doi.org/10.7554/eLife.31812.008
**Figure supplement 1.** The expression level of LAR and Ptp69D in the fly strain used in *Figure 3*.
DOI: https://doi.org/10.7554/eLife.31812.007

to the medulla. ΔD2 had less capability to rescue the phenotype than 2xCS, despite having a normal D1 phosphatase domain. Therefore, the ability of LAR to rescue the R7 stabilization phenotype is not necessarily dependent on phosphatase activity, and also relies on putative downstream interacting domain(s) residing in the proximity of D2.

On the other hand, *Ptp69D* constructs lacking catalytic activity of the first phosphatase domain DA1 did rescue the double mutant to the same extent as intact *Ptp69D* (*Figure 4I, K and N*) (*Garrity et al., 1999*; *Sun et al., 2001*). However, when both DA1 and DA2 catalytic activities were lost, Ptp69D did not rescue the R7 phenotype to any extent, as with the case for *Ptp69D* mutants

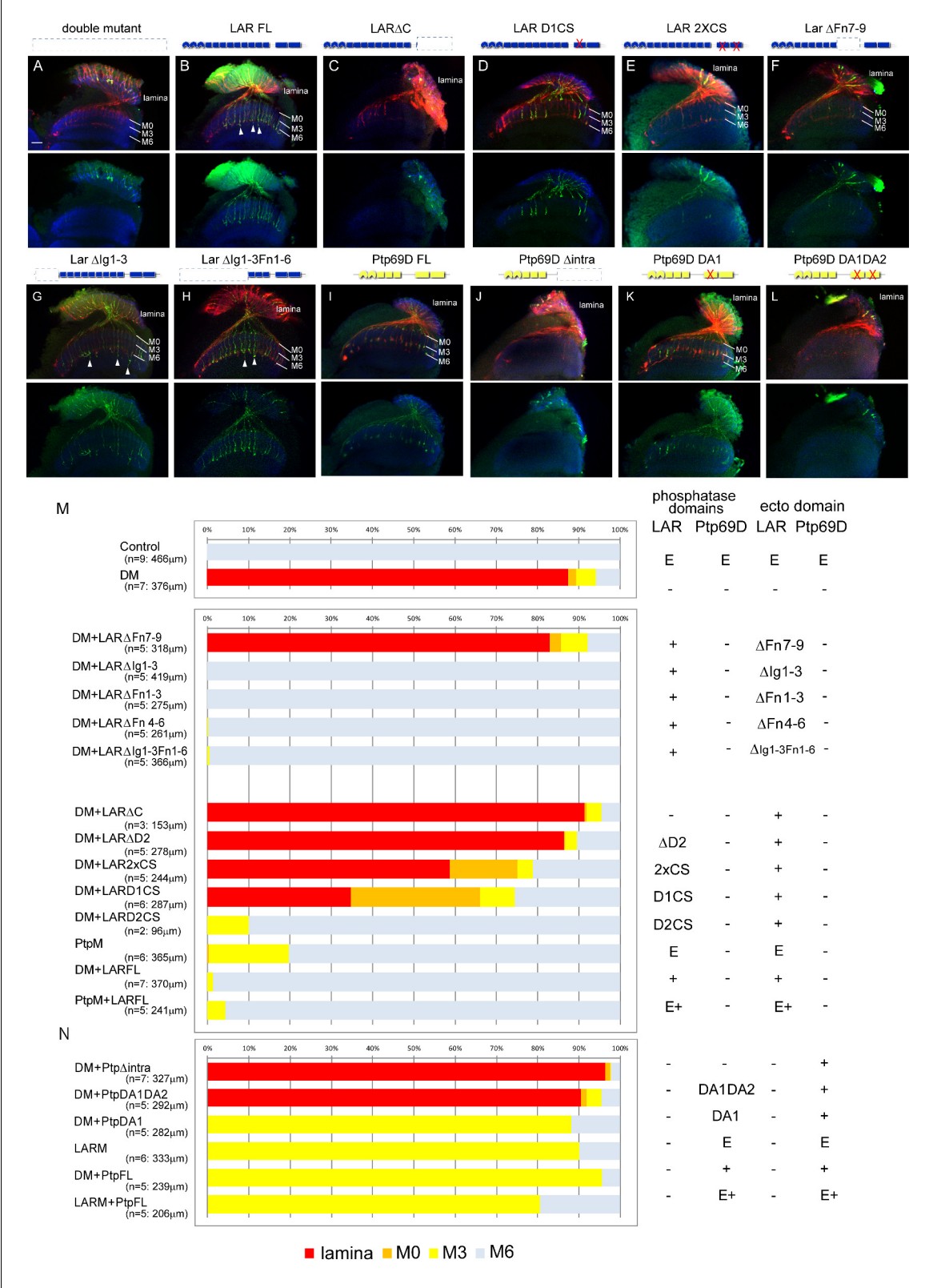

**Figure 4.** Specificity of RPTP signaling in R7 stabilization. (**A–L**) Horizontal images of the medulla of *LAR, Ptp69D* adult double mutant that also harbored the indicated transgene expressed using the GMR-Gal4 driver. Photoreceptor axons are labeled with mAb24B10 (red), R7 photoreceptor axons with Rh4-mCD8GFP (green), and the medulla layer with an antibody to N-Cadherin (blue). Medulla layers are indicated with white lines. R7 axon retraction in the *Lar, Ptp69D* double mutant eye (**A**) is almost completely rescued with full length Lar (LARFL), (**B**) but this is not the case if the

*Figure 4 continued*

intracellular domain (LARΔC) (**C**) or the extracellular FNIII 7–9 domains are deleted (LARΔFn7-9) (**F**). Lar constructs without activity from either the first phosphatase domain (LARD1CS) (**D**) or both phosphatase domains (LAR2XCS) (**E**) conveyed partial rescue. LAR constructs without extracellular Ig1-3 domains fully rescued R7 phenotype (**G**), and presence of extracellular FNIII 7–9 (LARΔIg1-3ΔFn1-6) was sufficient to rescue the double mutant phenotype (**H**). Notably, the constructs without Ig domains caused some R7 axons to overextend beyond M6 layer (arrowheads in G and H). Expressing either full length Ptp69D (PTP69D FL) (**I**) and Ptp69D lacking the first phosphatase domain (Ptp69D DA1) (**K**) rescued R7 retraction in *Lar, Ptp69D* double mutant axons to the *Lar* single mutant levels, but with termination in layer M3. Expression of Ptp69D lacking the intracellular domain (Ptp69DΔintra) (**J**) or both phosphatase activity (Ptp69D DA1DA2) (**L**) did not rescue the double mutant phenotype at all. (**M, N**) Quantification of R7 terminations in each layer for each of the indicated LAR (**M**) or Ptp69D (**N**) rescue transgenes. Abbreviations: DM stands for L*ar, Ptp69D* double mutant, PtpM for *Ptp69D* mutant, and LARM for *Lar* mutant. The transgenes of Ptp69D are abbreviated as 'Ptp'. On the right side of the graph, composition of the intracellular phosphatase domains and ectodomain are indicated. 'E' stands for endogenous protein and '+' represents overexpression by GMR-Gal4. The number of axons terminating inside the lamina was estimated as in *Figure 1*. Scale bar: 20 μm.

DOI: https://doi.org/10.7554/eLife.31812.009

The following source data and figure supplements are available for figure 4:

**Source data 1.** Excel file compiling source data for the *Figure 4M and N*.
DOI: https://doi.org/10.7554/eLife.31812.012
**Figure supplement 1.** The expression level of LAR and Ptp69D transgenes used in *Figure 4*.
DOI: https://doi.org/10.7554/eLife.31812.010
**Figure supplement 2.** Layer-specific binding of LAR and Ptp69D extracellular domain.
DOI: https://doi.org/10.7554/eLife.31812.011

lacking the entire cytoplasmic domain (*Figure 4J, L and N*). These results are consistent with previous reports (*Garrity et al., 1999*; *Hofmeyer and Treisman, 2009*; *Sun et al., 2001*) that unlike LAR intracellular domains, phosphatase activity is essential for Ptp69D signaling. We have assessed the expression level of the rescuing transgenes using antibody staining against larval eye discs (*Figure 4—figure supplement 1*). The transgene expression was monitored to show that they are all expressed above the endogenous level, in considerably equivalent and sufficient manner. This indicates that the rescuing ability is not correlated with the expression level of the transgenes, but rather with the protein domains and the mutations. Altogether, we have now successfully demonstrated that the degree of intracellular signaling plays a key role in layer termination.

## Different specificities in the LAR versus Ptp69D ectodomains contribute to the choice between layer M3 versus M6 stabilization

Despite LAR and Ptp69D functions being highly redundant, we noticed some differences between their phenotypes in the rescue experiments. Interestingly, in LAR rescue, the R7 axons that remained inside the medulla mostly stabilized at layer M6 or M0, but very few at layer M3 (*Figure 4M*), which stands in sharp contrast with the *LAR* mutant, where almost all R7 axon terminals stabilized in layer M3. This suggests that the LAR preferably directs R7 axon stabilization in layer M6, and to some extent M0.

In contrast, R7 axons in full length unmodified Ptp69D-rescued double mutants stabilized only in layer M3 (*Figure 4N*). Moreover, R7 axons were not stabilized in layer M6 even at the highest levels of Ptp69D expression (*Figure 4N*: LARM+PtpFL). These results indicate that Ptp69D alone does not have the ability to stabilize R7 axons to their proper layer M6, but does prevent R7 axons from retracting back to the lamina through adhesion in layer M3. Given that common domain accounting for this preference seems to reside in the ectodomains of LAR and Ptp69D (*Figure 4M and N*), we asked whether the expression pattern of potential ligands differs between LAR and Ptp69D. To detect ligand expression profile for each protein, we incubated the sectioned mid-pupal brain with an alkaline phosphatase (AP)-conjugated ectodomain of LAR and Ptp69D, a technique which is often used in embryonic nervous systems to detect the anatomical profile of ligand expression (*Fox and Zinn, 2005*). If the ectodomain binds to a potential ligand, ligand distribution among the medulla layers can be visualized through AP staining. Consistent with the genetic experiments, we observed a difference in ectodomain staining between LAR and Ptp69D: LAR ectodomain staining was mainly found in layers M0, M3, and M6 layers, while vague staining of the Ptp69D ectodomain was observed in layer M3 layer with no positive signal in layer M6 (*Figure 4—figure supplement 2*).

Taken together, these results demonstrate that LAR has a preference for directing R7 axons stabilization to layer M6, whereas Ptp69D selectively drives stabilization to M3. Both findings can be at least in part explained by differences in the localization pattern of putative LAR versus Ptp69D ligands.

## LAR and Ptp69D can determine the stabilizing layer of R8 axons

We next sought to determine if LAR and/or Ptp69D can also direct R8 axon stabilization in layer M3 (*Figure 5*, *Figure 5—figure supplement 1*). In *LAR, Ptp69D* double mutants, R8 axons terminated at the medulla surface M0 or between layers M1 and M3 (*Figure 5B and G*). We found that either LAR or Ptp69D expression in the double mutant restored R8 axon targeting (*Figure 5C, E and G*). We further found that LAR phosphatase activity was dispensable, whereas that of Ptp69D was essential for proper M3 targeting of R8 axons (*Figure 5D, F and G*). Furthermore, the overexpression of LAR in the double mutants resulted in some R8 axons extending past layer M3, and instead stabilizing in M6. The ratio of R8 axons terminating in layer M6 correlated with LAR cytoplasmic activity (*Figure 5C, D and G*). However, the mere overexpression of LAR did not cause overextension, likely due to M3 adhesive properties driven by Ptp69D (*Figure 5—figure supplement 1*). Based on these observations, we conclude that R8 axons follow similar rules as R7 axons in achieving their proper position. In each case, LAR stabilizes axons in layer M6, while Ptp69D stabilizes axons in layer M3.

## The LAR ectodomain can signal through the Ptp69D cytoplasmic domain

If activity of LAR and Ptp69D is driven by upstream ligands, it suggests that cytoplasmic deletions may behave as dominant negatives in some circumstances. This hypothesis led us to test whether LAR and Ptp69D lacking their intracellular domains (LARΔC and Ptp69DΔintra, respectively) sequestrate their putative ligand(s), thereby acting as dominant negatives (*Figure 6*). The overexpression of LARΔC or Ptp69DΔintra, as well as full length LAR and Ptp69D in the wild type background, did not show any dominant effects (*Figure 6G*, *Figure 6—figure supplement 1*). However, we found that LARΔC rescued the *LAR* mutant phenotype to some extent, as approximately half of R7 axons were targeted correctly to layer M6 (*Figure 6B and G*). In contrast, LARΔC enhanced the *Ptp69D* mutant phenotype with half of the R7 axons terminating in shallower layers (*Figure 6E and G*). When we expressed Ptp69DΔintra, both the *LAR* and *Ptp69D* mutant phenotypes were strongly enhanced (*Figure 6C, F and G*). In particular, the *LAR* mutant expressing Ptp69DΔintra showed as strong a phenotype as the double mutant (*Figure 6C and G*). These observations suggest that although LARΔC acts as a dominant negative, it can transduce signals directing R7 axons to layer M6, probably through Ptp69D activity as has been indicated previously (*Hofmeyer and Treisman, 2009*; *Maurel-Zaffran et al., 2001*). On the other hand, Ptp69DΔintra inhibited LAR function, indicating there is a common ligand for layer M6 targeting. We assume that Ptp69D can bind to this common ligand only when LAR is present. In addition, LAR must have the ligand that binds to R7 axons in M0 as well. Finally, our results indicate that Ptp69D binds to a selective ligand when R7 axons retract from layer M6 and that terminals stabilize in the temporary M3 layer in the absence of LAR.

## Redundant function of LAR and Ptp69D is cell-autonomous

To further elucidate the LAR and Ptp69D overlapping function, we used the GMR-FLP MARCM system (*Lee et al., 2001*; *Lee and Luo, 1999*) combined with RNAi (see Methods) to create single R7 axons of double-mutants (*Figure 7A–C*). In the control animals, we observed 57 R7 axons in 265 µm depth innervating the medulla and 95% of them terminated at the correct layer M6. As we could not directly count the R7 axons terminating inside the lamina, because some of R1 and R6 axons that normally target lamina were also labeled, we counted the R7 axons innervating medulla and compared it with the control animal. In the control animal, 21.5 R7 axons per 100 µm stack innervated the medulla, whereas only 7.4 axons did so with the double mutant clones, suggesting the cell-autonomous function of the RPTPs (*Figure 7C*). As we applied the cMARCM system (*Tomasi et al., 2008*), the WT axons were labeled with GMR-mCD8-Kusabira Orange (GMR-mCD8KO). Thus, the lack of KO signal indicates that the axon is a GFP positive clone. We thereby quantified the gaps in M6 terminating axons of GMR-mCD8KO (arrowheads in *Figure 7A and B*) and confirmed that the number of the double mutant R7 clones was comparable with the control (control: 22.3/100 µm,

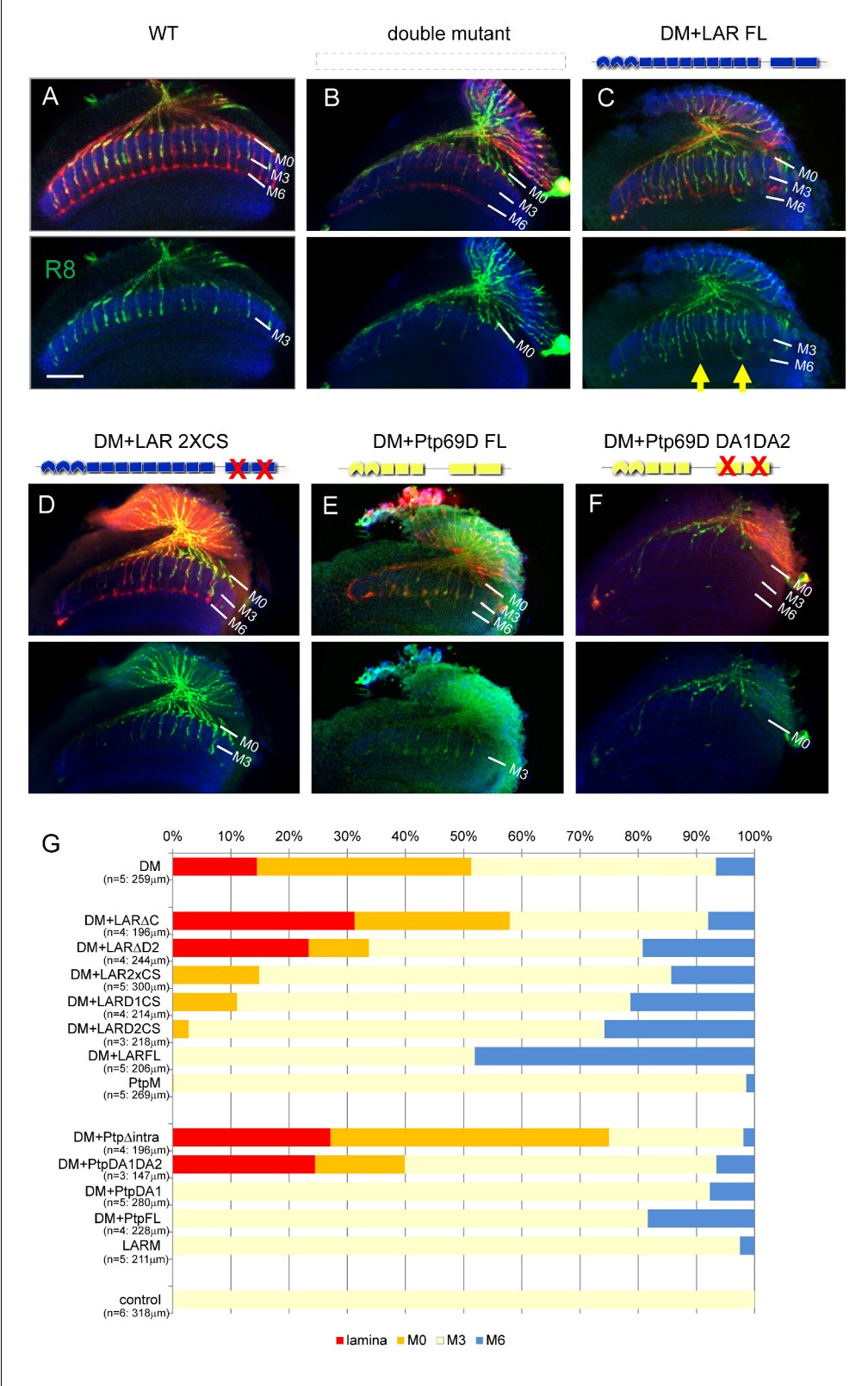

**Figure 5.** Specificity of RPTP signaling in R8 axons. (**A–F**) Horizontal images of adult medulla of *LAR, Ptp69D* double mutants also harboring the indicated transgene expressed under the GMR-Gal4 driver. Photoreceptor axons were labeled with mAb24B10 (red), R8 photoreceptor axons with Rh6-mCD8GFP (green), and the medulla layer with antibodies to N-Cadherin (blue). Medulla layers are indicated with white lines. The R8 axons in control flies terminated in M3 layer (**A**). In *Lar, Ptp69D* double mutant, more than half of the R8 axons terminated at the surface of the medulla M0 or the

*Figure 5 continued on next page*

*Figure 5 continued*

lamina (**B**). The R8 axon defects in double mutant were rescued with full length Lar (LARFL) with some axons overextending to layer M6 (arrows) (**C**). Lar constructs lacking phosphatase activity (LAR2XCS) rescued the double mutant phenotype, but do not cause axon overextension as the LARFL construct did (**D**). Full length Ptp69D (PTP69DFL) almost completely rescues the double mutant phenotype (**E**), but Ptp69D lacking phosphatase activity (Ptp69D DA1DA2) was incapable of rescue (**F**). (**G**) Quantification of R8 termination in each layer for the indicated LAR or Ptp69D rescue transgenes. Genotype descriptions are the same as in *Figure 4*. The number of axons terminating inside the lamina was estimated. Scale bar: 20 µm.

DOI: https://doi.org/10.7554/eLife.31812.013

The following source data and figure supplement are available for figure 5:

**Source data 1.** Excel file compiling source data for the *Figure 5G*.
DOI: https://doi.org/10.7554/eLife.31812.015
**Figure supplement 1.** Overexpression of LAR and Ptp69D has no effect on R8 axons.
DOI: https://doi.org/10.7554/eLife.31812.014

DM: 27.4/100 µm). It is likely that higher frequency of medulla innervation in GMR-FLP clone compared to the whole eye clone was due to the late timing of generation of the clones, thereby resulting in longer perdurance of the protein.

To test their cell-autonomy in complementary conditions, we also expressed the LAR transgene only in R7 or R8 cells using the specific Gal4 driver (*Figure 7—figure supplement 1*) in the double mutant to rescue the phenotype in cell-specific manner (*Figure 7D–I*). The R7 expression of LAR in double mutants showed that 77% of the R7 axons innervated the medulla (*Figure 7D–F*), and in the R8-specific rescue, 80% of R8 axons terminated at correct layer M3 (*Figure 7G–I*).

Taken together, these results indicate that overlapping function of LAR and Ptp69D in the photoreceptor neurons for axonal stabilization is cell autonomous.

## Abl and Trio operate downstream of LAR and Ptp69D in R7 axon stabilization

Genes including *Trio*, *Abl*, and *Ena* have been reported in previous work to be downstream targets of LAR and/or Ptp69D (*Newsome et al., 2000a*; *Wills et al., 1999*). However, as these studies focused only on the choice of R7 axons to stabilize at layer M3 versus M6, we sought to determine if these downstream components also function in preventing lamina retraction. To investigate what molecules act downstream of LAR and Ptp69D in R7 axons, we genetically tested the effects of Trio, Abl, and Ena upon the *LAR, Ptp69D* double mutant phenotypes (*Figure 8*, *Figure 8—figure supplement 1*). We first generated mutants with double *LAR, Ptp69D* RNAi knockdown genes under the control of GMR-Gal4. After RNAi knockdown, 35% of the R7 axons stopped at M0 and only 2% targeted deeper layers (*Figure 8A and G*). In this background, we then modulated the dosage of Abl, Ena, and Trio. After reducing Abl expression by RNAi, the ratio of R7 axons extending past layer M0 increased to ~13%, suggesting that Abl antagonizes the function of LAR and Ptp69D (*Figure 8B and G*). Contrastingly, *Trio* RNAi enhanced the phenotype, as the ratio of R7 axons stopping at M0 decreased to 10% (*Figure 8E and G*). *Ena* RNAi alone did not have an effect in the double *LAR, Ptp69D* RNAi-knockdown background (*Figure 8D and G*). When the dosage of these three molecules was increased using UAS transgenes, the opposite effects with respect to the RNAi-knockdown phenotypes were observed. Overexpression of Abl enhanced the *LAR, Ptp69D* double RNAi phenotype with only 7% of R7 axons innervating the medulla (*Figure 8C and G*), while expression of Trio suppressed the phenotype and more than 60% of axons innervated the medulla, reaching layers M3 or M6 (*Figure 8F and G*). As a control, we confirmed that overexpression of these genes in a wild type background did not have any effect (*Figure 8—figure supplement 1*). Taken together, our results indicate that Abl and Trio interact with the LAR and Ptp69D intracellular domains, acting as key components of a common downstream pathway directing the stabilization of R7 axons. Thus, the unique properties of LAR versus Ptp69D are conveyed by properties of their extracellular domains. In other words, in the determination of the final stabilizing layers, the activation signals in each of the layers utilize the same downstream pathway, but ligand expression and binding appears to be different (*Figure 9*).

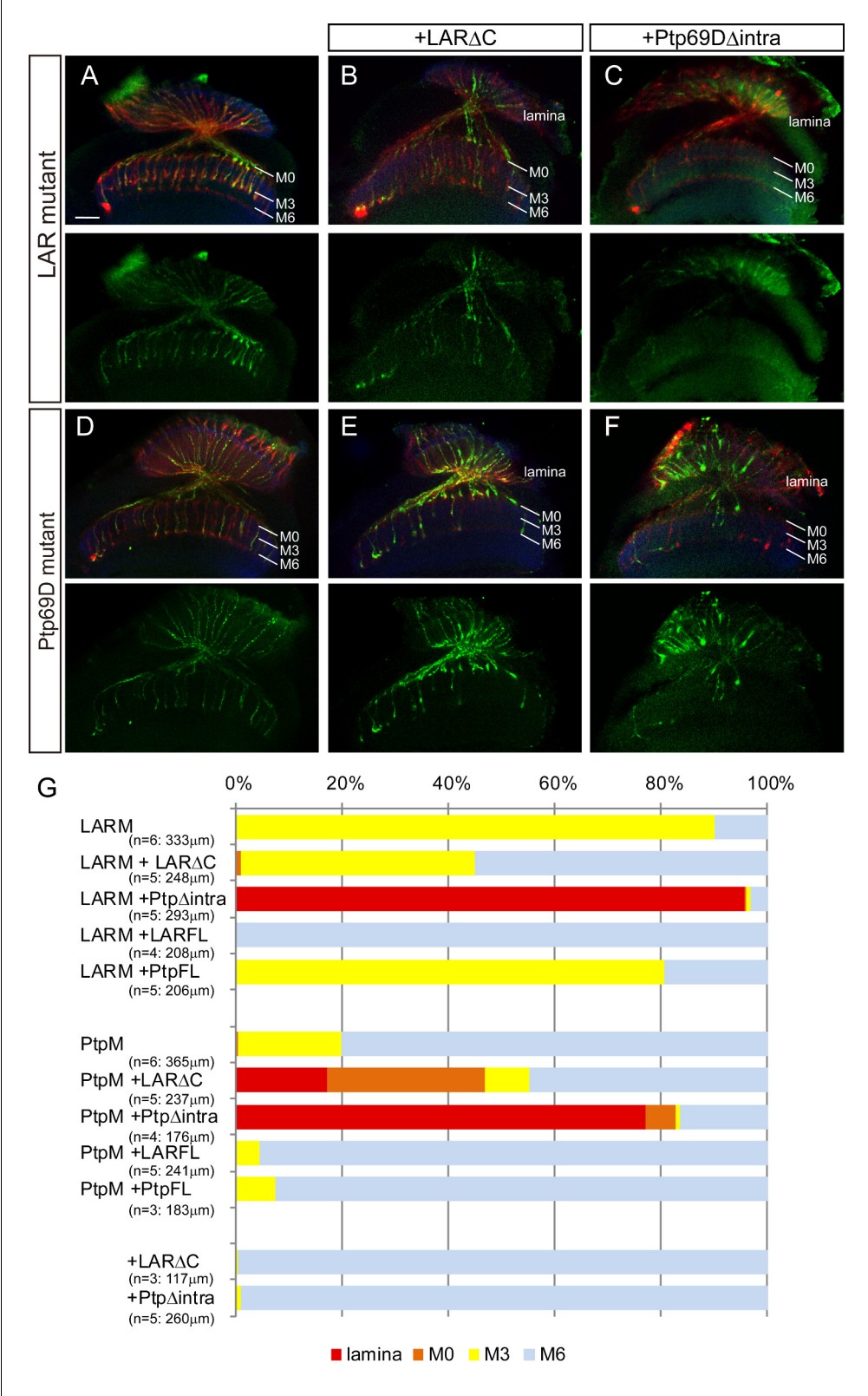

**Figure 6.** Specificity of RPTP ectodomains in R7 axons. (A–F) Horizontal images of adult medulla of *LAR* (A–C) or *Ptp69D* (D–F) single mutants carrying the indicated transgene expressed by the GMR-Gal4. Photoreceptor axons are labeled with mAb24B10 (red), R7 photoreceptor axons with Rh4-mCD8GFP (green), and the medulla layer with an antibody to N-Cadherin (blue). Medulla layers are indicated with white lines. A total of 90% of the R7 axons terminated in layer M3 in the *LAR* mutant eye (A). Half of the R7 axons terminated at proper M6 layer when the *LAR* mutant was rescued with LAR

*Figure 6 continued on next page*

*Figure 6 continued*

transgene lacking intracellular domain (LARΔC) (B), while Ptp69D transgene lacking intracellular domain (Ptp69DΔintra) enhances the phenotype and almost all the R7 axons terminated in the lamina (C). A total of 20% of the R7 axons terminated in layer M3 in the *Ptp69D* mutant eye (D). Both LAR transgene lacking intracellular domain (LARΔC) (E) and Ptp69D transgene lacking intracellular domain (Ptp69DΔintra) (F) enhanced the phenotype of *Ptp69D* mutant. (G) Quantification of R7 terminations in each in the indicated mutants rescued with LAR or Ptp69D transgenes. Note that neither LARΔC nor Ptp69DΔintra had a dominant effect. Genotype descriptions are the same as in *Figure 4*. The number of axons terminating inside lamina was estimated as in *Figure 1*. Scale bar: 20 μm.

DOI: https://doi.org/10.7554/eLife.31812.016

The following source data and figure supplement are available for figure 6:

**Source data 1.** Excel file compiling source data for the *Figure 6G*.

DOI: https://doi.org/10.7554/eLife.31812.018

**Figure supplement 1.** Overexpression of LAR and Ptp69D.

DOI: https://doi.org/10.7554/eLife.31812.017

## Discussion

### Two principles of axon stabilization regulated by RPTPs

Here, we show that R7 axons lacking the RPTPs LAR and Ptp69D failed to innervate the medulla and instead terminated inside the lamina (*Figure 1*). By observing axon extension in the medulla at several developmental stages, we revealed that the double mutant R7 axons extended normally at the beginning of the pupal stage, but then gradually retracted from their correct temporary layer and terminated in the wrong layer (*Figure 2*). These data demonstrate that LAR and Ptp69D play a common role in the stabilization of R7 axons inside the medulla. The simultaneous manipulation of the expression levels of each gene revealed two principles explaining their shared and common functions in axon guidance and stabilization. First, determination of the final termination layer of R7 axons primarily relied on the cumulative expression level of both LAR and Ptp69D (*Figure 3*). The amount of the cytoplasmic transduction was proportional to the depth of the final termination layer, regardless of the specific RPTP (LAR versus Ptp69D). Second, although LAR and Ptp69D share a common cytoplasmic pathway, thus explaining the high redundancy of the two molecules, unique properties in their ectodomains were responsible for the choice between layers M3 and M6 of the medulla (*Figure 9*). Together, these principles can explain the difference between the LAR versus Ptp69D single mutant phenotypes, and also explain why LAR can rescue Ptp69D function, but not vice versa.

However, an alternative explanation of the proportional relationship between cumulative LAR and Ptp69D expression levels and the depth of the stabilizing layer is that LAR and Ptp69D cytoplasmic activity is only required to stabilize the axon whenever the ligand of these receptors is bound. The LAR ectodomain exhibits preferential expression in layers M0 and M6, whereas the Ptp69D ectodomain attaches to layer M3 (*Figure 4—figure supplement 2*), where their respective ligands are presumably expressed (*Figure 9B*). This model is supported by the double mutant rescue experiment, in which ectopic expression of LAR variants with different cytoplasmic activity level in the double knockout background led to extensive axon termination at layers M6 or M0, but very rare termination in M3. On the other hand, double mutant R7 axons rescued by Ptp69D variants mostly terminated in layer M3 (*Figure 4*). Even a high-dosage overexpression of Ptp69D was insufficient to stabilize R7 axons in layer M6 in the absence of LAR (*Figure 6G*), indicating that the Ptp69D ectodomain is able to stabilize R7 axons only at layer M3, rather than in layer M6. Consistent with the findings for R7 axons, about half of R8 axons displayed ectopic extension and stabilization to layer M6 layer instead of layer M3 when LAR was overexpressed in the absence of Ptp69D, whereas Ptp69D overexpression did not generate a prominent phenotype (*Figure 5*). However, even in this latter scenario, some degree of correlation between layer-depth and RPTP activity must exist, as R7 axons require less LAR cytoplasmic activity for stabilization in layer M0 than in layer M6.

### Two models of axonal stabilization regulated by RPTPs

How mechanistically can less LAR and Ptp69D activity be required to stabilize an axon when it retracts back and becomes shorter? Our data are consistent with the notion that LAR and Ptp69D likely bind and activate adhesion molecules to stabilize the growth cone to the target site in a

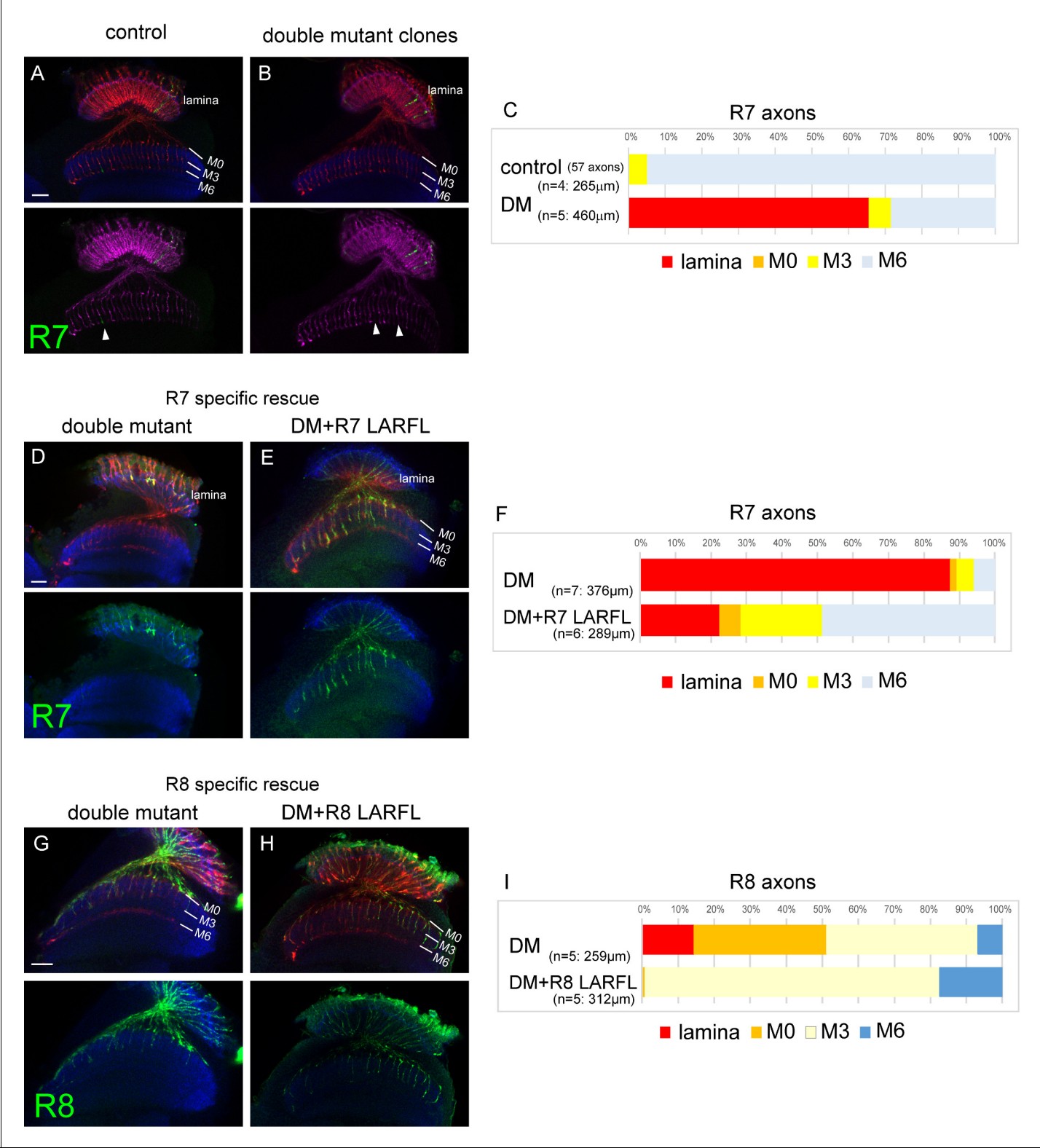

**Figure 7.** Cell-autonomous function of LAR and Ptp69D. (**A–C**) Single R7 clones of LAR, Ptp69D double mutant. Ptp69D mutation was introduced by creating the clone with MARCM system using GMR-FLP, whereas LAR expression was down-regulated by combining LAR heterozygous mutation with LAR-RNAi expressed only in the clones by GMR-Gal4. The detailed genotypes are the following: (A: control) GMR-FLP, UAS-mCD8GFP/+; GMR-Gal4/+; FRT80/tub-Gal80, GMR-mCD8KO, FRT80, (B: double mutant) GMR-FLP, UAS-mCD8GFP/+; LAR[2127], GMR-Gal4/UAS-LARRNAi; Ptp69D[D1689] FRT80/ tub-Gal80, GMR-mCD8KO, FRT80. (**A, B**) Horizontal images of adult medulla; mCD8GFP was used to label mutant R7 axons (green), whereas

*Figure 7 continued on next page*

*Figure 7 continued*

mCD8-Kusabira Orange (red) to complementarily label wild type photoreceptor cells. The medulla layer was visualized with an antibody to N-Cadherin (blue). In the control animals, single R7 axons labeled in green extending to M6 layer were observed (arrowhead in A). However, when double mutant clones were generated, some axon terminals in M6 layer were absent (arrowheads in B) indicating that double mutant R7 axons failed to innervate the medulla. (C) Quantification of R7 termination in each layer. The estimation of total number of R7 axons was made from the control medulla. 65% of the double mutant R7 axons terminated inside the lamina. (D, E) Horizontal images of the adult medulla of *LAR, ptp69D* double mutants carrying the LARFL transgene expressed by the R7-specific PM181-Gal4. Photoreceptor axons were labeled with mAb24B10 (red), R7 photoreceptor axons with Rh4-mCD8GFP (green), and the medulla layer with an antibody to N-Cadherin (blue). With R7-specific LAR expression in *LAR, ptp69D* double mutants, about half of the R7 axons terminated at the proper layer (M6) and 77% innervated the medulla (E). (F) Quantification of the R7 axons terminating in each layer. (G, H) Horizontal images of the adult medulla of *LAR, ptp69D* double mutants carrying the LARFL transgene expressed by the R8-specific 2–80–Gal4. Photoreceptor axons were labeled with mAb24B10 (red), R8 photoreceptor axons with Rh6-mCD8GFP (green), and the medulla layer with an antibody to N-Cadherin (blue). With R8-specific LAR expression in *LAR, ptp69D* double mutants, about 80% of the R8 axons terminated at the proper layer (M3) and 17% overextended to the M6 layer. (I) Quantification of the R8 axons terminating in each layer. The number of axons terminating inside the lamina was estimated. Scale bars: 20 μm.

DOI: https://doi.org/10.7554/eLife.31812.019

The following source data and figure supplement are available for figure 7:

**Source data 1.** Excel file compiling source data for the *Figure 7C and F* and 7I.
DOI: https://doi.org/10.7554/eLife.31812.021
**Figure supplement 1.** Specificity of the PM181-Gal4 and 2–80 Gal4.
DOI: https://doi.org/10.7554/eLife.31812.020

contact mediated manner. One scenario might be that the ligands form a gradient that is higher along the proximal part of the photoreceptor axons. Thus, a lower amount of receptor protein can still generate a sufficient amount of stabilizing signal. This scenario is unlikely, however, because the ligand has to be expressed in higher doses where normally it is not functionally required. The second, and a more likely scenario is that the lower amounts of RPTP signaling are required when the axon is shorter and the growth cone is closer to the cell body. It is important to note that the RPTP stabilization signal seems to be required in mature axons but not young growing axons. In this scenario, layer-enriched expression of RPTP ligands would ensure that axons that had grown into the medulla would notice a stabilizing ligand at their maximum length. This mechanism would serve as a regulatory check on axons that failed to stabilize at a certain target layer by allowing them to stabilize at more proximal layers. This process offers both a universal and economical model by which striped expression of a limited number of ligands can shape a relay of neuronal connections comprised of constant length axons connected from the surface to the inner region of the brain. In addition, the system requires only a few guidance molecules to guide axons in a certain direction, rather than an array of specific molecules to each targeting layer and stabilizing specific class of neurons.

## Biphasic properties of axonal growth

Given the observation that double mutant photoreceptors extend their axons normally until 24hrAPF, and then suddenly started to retract them around 40hrAPF, there seems to be a biphasic pattern to axonal growth. First, R7 axons grow by default, and seem to neither require nor recognize stabilizing RPTP signaling. In the later stage of pupal development, R7 axons retract by default, and require RPTP signaling for stabilization. The difference between these two phases might be attributable to changes in molecular properties inside axons, in particular the dynamics of the actin polymerization and microtubule organization. One of the triggers that can cause axon maturation and terminal targeting can be the formation of synapses, and/or firing of action potentials. To test whether double mutant R7 retraction can be rescued by neuronal activity, we expressed TrpA1 channel and elevated the temperature to activate the firing of R7 neurons. However, they still retracted back to the lamina, suggesting that activity is not sufficient for suppressing axon retraction at least in RPTP mutant backgrounds (*Figure 9—figure supplement 1*).

## Formation of complexes between LAR and Ptp69D

Previous studies from Jessica Treisman's research group have provided ample evidence that LAR and Ptp69D form complexes with each other. Treisman and colleagues analyzed R7 axons in *LAR* mutants and found that LAR phosphatase activity is dispensable for stabilization in layer M6 because

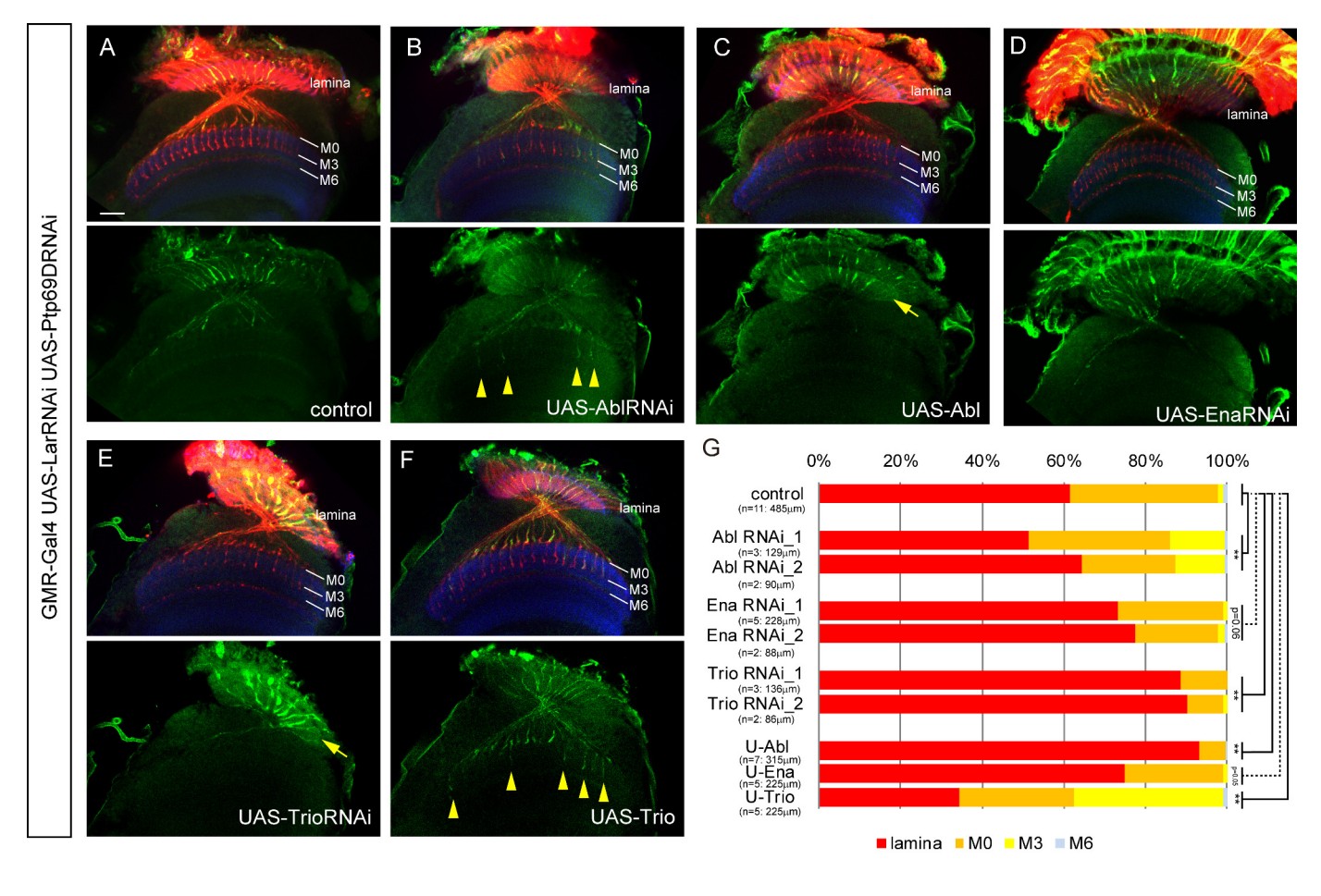

**Figure 8.** Genetic interactions between LAR, Ptp69D and Abl, Ena, and Trio in R7 targeting. (A–F) The expression of Abl, Ena, and Trio was downregulated using RNAi or upregulated by overexpression of transgenes in *LAR, Ptp69D* double RNAi background. All the RNAi and transgenes were expressed under the GMR-Gal4 driver. Photoreceptor axons are labeled with mAb24B10 (red), R7 photoreceptor axons with Rh4-mCD8GFP (green), and the medulla layers with an antibody to N-Cadherin (blue). The medulla layers are indicated with white lines. Around 60% of R7 axons expressing the *LAR, Ptp69D* double RNAi terminated inside the lamina. The phenotype is suppressed by downregulation of Abl (B) or upregulation of Trio (F) with some R7 axons terminating in layer M3 (arrowheads). By contrast, upregulation of Abl (C) or downregulation of Trio (E) enhanced the phenotype with most R7 axons terminating in the lamina (arrows). Ena had no significant effect on the phenotype (p=0.06) upon both downregulation (D) and upregulation (not shown). (G) Quantification of R7 terminations in each layer in the control *LAR, Ptp69D* double RNAi line with and without each indicated RNAi or overexpression transgene of Abl, Ena, and Trio. RNAi_1 and RNAi_2 refer to the two different RNAi lines used for each gene. '**' indicates statistical significance at the p<0.01 level using chi-square test. The number of axons terminating inside the lamina was estimated. Scale bar: 20 µm.

DOI: https://doi.org/10.7554/eLife.31812.022

The following source data and figure supplement are available for figure 8:

**Source data 1.** Excel file compiling source data for the *Figure 8G*.
DOI: https://doi.org/10.7554/eLife.31812.024

**Figure supplement 1.** RNAi and overexpression of Abl, Ena and Trio.
DOI: https://doi.org/10.7554/eLife.31812.023

of the catalytic activity of Ptp69D, and determined that LAR phosphatase activity is required only in the absence of Ptp69D (*Hofmeyer and Treisman, 2009*).

In our study, we labeled R7 axons specifically and aimed to dissect the function of the ectodomain and cytoplasmic domain of LAR and Ptp69D. We found that LAR lacking phosphatase activity still stabilized R7 axons inside the medulla to some extent, even without Ptp69D (*Figure 4E and M*). When LAR lacks the entire cytoplasmic domain (LARΔC), it could not rescue the double mutant at all

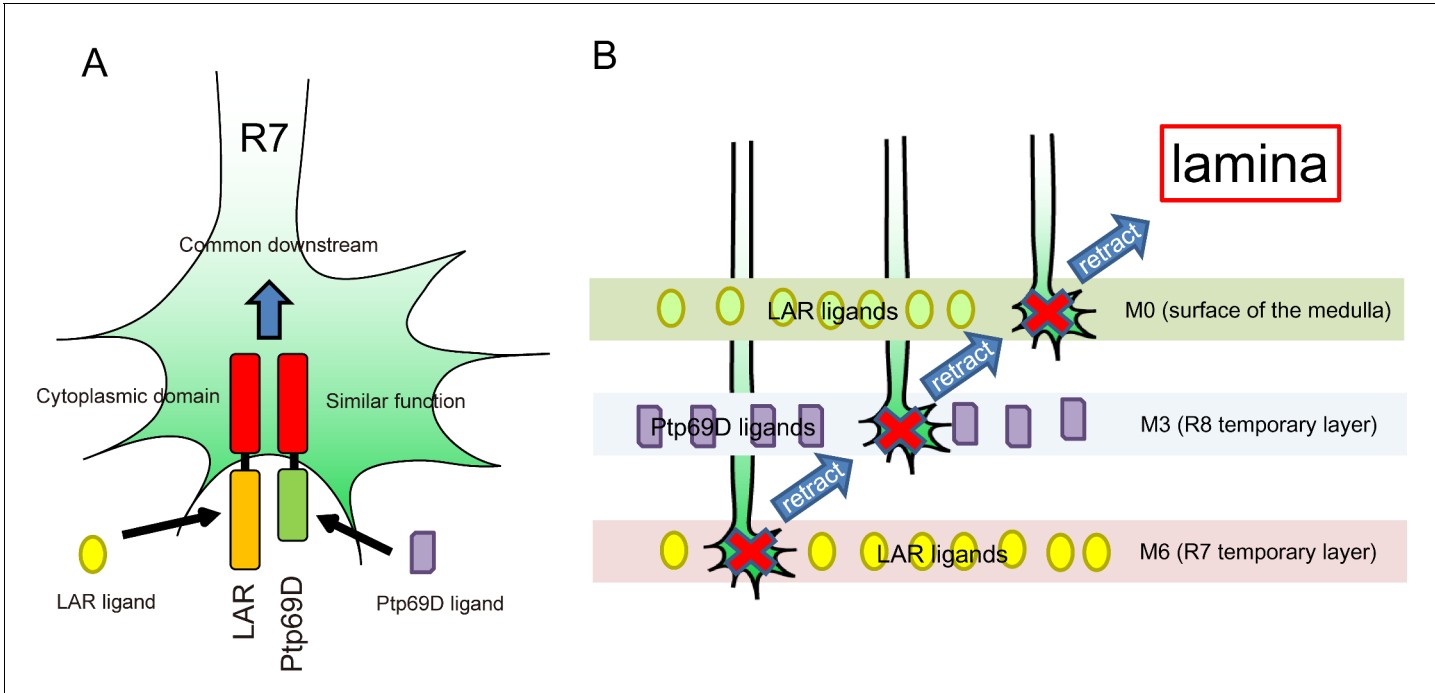

**Figure 9.** A model for common and distinct functions of LAR and Ptp69D in R7Axons. **(A)** Both LAR and Ptp69D are expressed in R7 axon terminals. Their ectodomains bind to distinct ligands whereas intracellular domains share a common downstream pathway. **(B)** Around 24hrAPF, R7 axons reach the R7 temporary layer, which becomes M6. When they fail to detect Lar ligands, however, they start to retract. While retracting, the terminals of R7 axons pass by the M3 layer. If they detect Ptp69D ligands, they stabilize at M3, but if unbound by ligands they keep retracting. The next layer is M0 where LAR ligands appear to also be present. Here, lower signaling levels, even without phosphatase activity, can stabilize axons. If axon stabilization fails across M0, M3, and M6, R7 axons retract to the lamina.

DOI: https://doi.org/10.7554/eLife.31812.025

The following figure supplement is available for figure 9:

**Figure supplement 1.** R7 axon stabilization is not dependent on neuronal activity.
DOI: https://doi.org/10.7554/eLife.31812.026

and R7 axons retracted to the lamina (**Figure 4C and M**). On the contrary, when LARΔC is combined with full length Ptp69D, more than half of the R7 axons terminated at the proper layer M6 (**Figure 6B and G**), although Ptp69D alone could only stabilize R7 axons in layer M3 (**Figure 4I and N**). A previous study showed similar results concluding that LAR has non-autonomous functions in R8 axons, serving as a ligand for an unknown receptor in R7 axons to promote targeting to layer M6 (**Maurel-Zaffran et al., 2001**). Based on our results, this seems unlikely because LARΔC enhanced the *Ptp69D* null mutant phenotype (**Figure 6E and G**) and had no function in the double mutant (**Figure 4C and M**). Our data instead support the idea that LAR and Ptp69D create a complex, and that upon binding of LAR to its ligand in layer M6 signals for stabilization are transduced through the phosphatase domains of Ptp69D even in the absence of LAR cytoplasmic domains. In contrast, when the intracellular domain of Ptp69D is deleted (Ptp69DΔintra) and then expressed in the *Ptp69D* mutant, the Ptp69D ectodomain alone was sufficient to prevent LAR effects and most R7 axons retracted to the lamina. Similarly, Ptp69DΔintra also suppressed Ptp69D function in *LAR* mutants. Based on these rescue experiment data, Ptp69D probably creates a complex with LAR that binds to unknown LAR ligand in layer M6. When they are in the complex, the stabilizing signal in layer M6 is likely transduced through the Ptp69D intracellular domain.

Although Sdc and Dlp are thought to be ligands of LAR that promote synapse formation in the neuromuscular system (**Fox and Zinn, 2005**; **Johnson et al., 2006**), it is unlikely that they serve as ligands in the visual system. Some defects were observed in photoreceptor axon pathfinding in both *sdc* and *dlp* mutants, but R7 axons did not show any significant retraction at 40hrAPF (**Rawson et al., 2005**). Moreover, Sdc and Dlp bind to the extracellular Ig domains of LAR

(*Johnson et al., 2006*), but the transgene lacking three Ig domains can rescue R7 targeting in the *LAR* mutant (*Hofmeyer and Treisman, 2009*) as well as in the double mutant (*Figure 4G and M*), and FnIII domains of LAR are essential to stabilize R7 axons inside the medulla (*Figure 4F, H and M*).

Then what is the ligand of LAR? We consider N-Cadherin (CadN) to be one of the candidates. The phenotype of R7 axons lacking CadN resembles the *LAR* mutant with axon termination in layer M3. CadN is required to reach and remain in the R7 temporary layer at the early stage of pupal formation, whereas the retraction of R7 axons in *LAR* mutant starts at a later stage (*Ting et al., 2005*). Detailed analysis of photoreceptor filopodial dynamics by the live imaging revealed that CadN deficient R7 axons initially reached the correct target layer but failed to stabilize there. This observation suggests that CadN is not a guidance cue but a stabilization factor (*Özel et al., 2015*). In addition to their similar functions in R7 axon stabilization, *Drosophila* CadN and LAR physically interact in vitro (*Prakash et al., 2009*). It was also reported that the mammalian N-Cadherin is an in vivo substrate of the mammalian protein tyrosine phosphatase sigma (PTPσ), which is an orthologue of *Drosophila* LAR. Dephosphorylation of N-Cadherin by PTPσ increase adhesiveness and inhibits axon growth (*Siu et al., 2007*). It is thus highly conceivable that CadN in R7 axons and their target layers interact with the extracellular domain of the LAR-PTP69D complex and that phosphorylation status of CadN affects the adhesiveness between the R7 axon growth cone and the proper target layer, eventually resulting in axonal stabilization in the *Drosophila* visual system.

# Materials and methods

**Key resource table**

| Reagent type (species) or resource | Designation | Source or reference | Identifiers |
|---|---|---|---|
| gene (Drosophila) | LAR | NA | |
| | Ptp69D | NA | |
| strain, strain background (Drosophila) | *D. melanogaster*: ey-FLP2 | (*Newsome et al., 2000a*) | BDRC5580 |
| | *D. melanogaster*: Ptp69D[D1689] | (*Newsome et al., 2000a*) | DGRC109897 |
| | *D. melanogaster*: LAR[2127] | (*Maurel-Zaffran et al., 2001*) | BDRC63796 |
| | *D. melanogaster*: UAS-LAR | (*Maurel-Zaffran et al., 2001*) | N/A |
| | *D. melanogaster*: UAS-LARΔC | (*Maurel-Zaffran et al., 2001*) | N/A |
| | *D. melanogaster*: UAS-LARΔIg1-3ΔFn1-6 | (*Hofmeyer and Treisman, 2009*) | N/A |
| | *D. melanogaster*: Rh4-mCD8GFP | (*Berger et al., 2001*) | N/A |
| | *D. melanogaster*: Rh6-mCD8GFP | (*Berger et al., 2001*) | N/A |
| | *D. melanogaster*: 20C11-FLP | (*Chen et al., 2014*) | BDRC63796 |
| | *D. melanogaster*: UAS-Trio | (*Newsome et al., 2000b*) | N/A |
| | *D. melanogaster*: UAS-Abl | (*Fogerty et al., 1999*) | N/A |
| | *D. melanogaster*: UAS-Ena | (*Wills et al., 1999*) | N/A |
| | *D. melanogaster*: UAS-FRT-stop-FRT-mcd8GFP | Bloomington Drosophila Stock Center | BDRC30032 |
| | *D. melanogaster*: tub-Gal80[ts] | Bloomington Drosophila Stock Center | BDRC7108 |
| | *D. melanogaster*: UAS-Lar.ΔIg123 | Bloomington Drosophila Stock Center | BDRC8586 |
| | *D. melanogaster*: UAS-Lar.ΔFn123 | Bloomington Drosophila Stock Center | BDRC8587 |
| | *D. melanogaster*: UAS-Lar.ΔFn456 | Bloomington Drosophila Stock Center | BDRC8588 |

*Continued on next page*

Continued

| Reagent type (species) or resource | Designation | Source or reference | Identifiers |
|---|---|---|---|
| | D. melanogaster: UAS-Lar.ΔFn789 | Bloomington Drosophila Stock Center | BDRC8589 |
| | D. melanogaster: UAS-Lar.ΔPTP-D2 | Bloomington Drosophila Stock Center | BDRC8590 |
| | D. melanogaster: UAS-Lar.C1638S | Bloomington Drosophila Stock Center | BDRC8591 |
| | D. melanogaster: UAS-Lar.C1929S | Bloomington Drosophila Stock Center | BDRC8592 |
| | D. melanogaster: UAS-Lar.CSX2 | Bloomington Drosophila Stock Center | BDRC8593 |
| | D. melanogaster: UAS-Ptp69D Δintra | KYOTO Stock Center | DGRC109088 |
| | D. melanogaster: UAS-Ptp69D DA1 | KYOTO Stock Center | DGRC109089 |
| | D. melanogaster: UAS- Ptp69D DA3(DA1DA2) | KYOTO Stock Center | DGRC109090 |
| | D. melanogaster: UAS-Ptp69D | KYOTO Stock Center | DGRC109091 |
| | D. melanogaster: UAS-LAR RNAi | Vienna Drosophila Resource Center | VDRC36269 |
| | D. melanogaster: UAS-ptp69D RNAi | Vienna Drosophila Resource Center | VDRC27090 |
| | D. melanogaster: UAS-abl RNAi | Bloomington Drosophila Stock Center | BDRC28325, 35327 |
| | D. melanogaster: UAS-ena RNAi | Bloomington Drosophila Stock Center | BDRC31582, 39034 |
| | D. melanogaster: UAS-trio RNAi | Bloomington Drosophila Stock Center | BDRC27732, 43549 |
| | D. melanogaster:<LAR < | This paper | Harvard Exelixis collection; e04149, e00822 |
| | D. melanogaster:<Ptp69D< | This paper | Harvard Exelixis collection; f03442, e00274 |
| antibody | mAb24B10 | Developmental Studies Hybridoma Bank | 24B10 |
| | rat antibody to CadN(N-Cad) | Developmental Studies Hybridoma Bank | DN-Ex #8 |
| | rat antibody to elav | Developmental Studies Hybridoma Bank | 7E8A10 |
| | mouse antibody to LAR | Developmental Studies Hybridoma Bank | 9D8 |
| | mouse antibody to Ptp69D | Developmental Studies Hybridoma Bank | 3 F11 |
| | GFP Tag Polyclonal Antibody, Alexa Fluor 488 | Life technologies | Cat #A-21311 |
| | Goat anti-Mouse IgG (H + L) Highly Cross-Adsorbed Secondary Antibody, Alexa Fluor 568 | Life technologies | Cat #A-11011 |
| | Goat anti-Rat IgG (H + L) Cross-Adsorbed Secondary Antibody, Alexa Fluor 633 | Life technologies | Cat #A-21094 |
| | Anti-Placental alkaline phosphatase (PLAP) | Abcam | Cat #ab118856 |

*Continued*

| Reagent type (species) or resource | Designation | Source or reference | Identifiers |
|---|---|---|---|
| | Goat Anti-Rabbit IgG H&L (Alexa Fluor 488) preabsorbed | Abcam | Cat #ab150085 |
| | rabbit antibody to HA | Abcam | Cat #ab9110 |
| chemical compound, drug | Paraformaldehyde 16% | Nisshin EM | Cat#15710 |
| | Agar powder (gelling temperature 30–31°C) | Nacalai tesque | Cat#01059–85 |
| | Vectashield mouting medium | Vector Laboratories | Funakoshi #H-1000 |
| software, algorithm | NIS-Elements AR | Nikon | |
| | Adobe Photoshop CS6 | Adobe | |

## Fly strains and genetics

Flies were kept in standard Drosophila media at 25°C, except for the experiment of experiment for manipulating expression level (27°C, 25°C, 23°C and 18°C) and temporal expression level shift (27°C and 18°C). To manipulate the expression level of two RPTPs simultaneously, *LAR, ptp69D* double RNAi was expressed by eye-specific GMR-Gal4 and RNAi expression was controlled using temperature sensitive Gal80$^{ts}$ at 23°C, 25°C, and 27°C. To achieve lower expression level, we also expressed *LAR, ptp69D* double RNAi in the back ground of LAR$^-$/+, ptp69D$^-$/+ and utilize the temperature dependency of Gal4 strength (25°C and 18°C). To perform temporal control of the expression level, *LAR* and *ptp69D* RNAi were expressed in photoreceptors by GMR-Gal4 controlled by temperature sensitive Gal80$^{ts}$ as above, and temperature was shifted from 27°C to 18°C or from 18°C to 27°C at indicated time point. The stages after puparium formation are determined according to (*Ashburner, 1989*). For the FLICK technique, two FRT lines flanking a gene to be knocked out were recombined onto the same chromosome. These FRT lines were obtained from Exelixis collection: LAR; e04149, e00822, ptp69D; f03442, e00274 (*Thibault et al., 2004*). A more detailed explanation about FLICK is provided in (*Hakeda-Suzuki et al., 2011*). LAR and Ptp69D FLICK flies are referred to as <LAR< and <Ptp69D< respectively. To create the double mutant R7 single clone, Ptp69D mutation was introduced by creating the clone with MARCM system using GMR-FLP (*Lee et al., 2001*; *Lee and Luo, 1999*), whereas LAR expression was down-regulated by combining LAR heterozygous mutation with LAR-RNAi expressed only in the clones by GMR-Gal4. The following fly stocks and mutant alleles were used: yw, eyFLP2, Ptp69D[D1689] (*Newsome et al., 2000a*), LAR[2127], UAS-LARFL, UAS-LARΔC, PM181-Gal4 (*Maurel-Zaffran et al., 2001*), UAS-LARΔIg1-3ΔFn1-6 (*Hofmeyer and Treisman, 2009*), Rh4-mCD8GFP, Rh6-mCD8GFP (*Berger et al., 2001*), 20C11-FLP (*Chen et al., 2014*), UAS-Trio (*Newsome et al., 2000b*), UAS-Abl (*Fogerty et al., 1999*), UAS-Ena (*Wills et al., 1999*), UAS-FRT-stop-FRT-mCD8GFP, tub-Gal80[ts], GMR-myrRFP, UAS-LAR* (#8589-#8593) transgenic lines were obtained from Bloomington stock center. UAS-ptp69D* (#109088-#109091) transgenic lines were obtained from Kyoto stock center. LAR RNAi (v36269) and ptp69D RNAi (v27090) are from Vienna stock center, Abl RNAi (#28325, #35327), Ena RNAi (#31582, #39034), Trio RNAi (#27732, #43549) lines are from Bloomington stock center. R8-specific driver 2–80 Gal4 is a Gal4 enhancer trap line on the X chromosome kindly provided by Tory Herman. R8- specific expression persists until mid-pupal stage. The specific genotypes utilized in this study are listed in *Supplementary file 1*.

## Immunohistochemistry and imaging

The experimental procedures for adult brain dissection, fixation and immunostaining as well as agarose section were as described previously (*Hakeda-Suzuki et al., 2011*). Larval eye-brain complexes were dissected in PBS and fixed in 4% paraformaldehyde in PBS for 30 min. The staining was performed in PBS with 0.1% Saponin for LAR antibody and with 0.1% Triton for Ptp69D antibody. The

following primary antibodies were used: mAb24B10 (1:50, DSHB), rat antibody to CadN (Ex#8, 1:50, DSHB), rat antibody to elav (7E8A10, 1:100 DSHB), mouse antibody to LAR (9D8, 1:20 DSHB), mouse antibody to Ptp69D (3F11, 1:20 DSBH), rabbit antibody to HA (Ab9110, 1:1000 Abcam) and rabbit antibody to GFP conjugated with Alexa488 (1:200, Life technologies). The secondary antibodies were Alexa488, Alexa568 or Alexa633-conjugated (1:400, Life technologies). Images were obtained with Nikon C2$^+$ confocal microscopes and processed with NIS-elements AR and Adobe Photoshop.

## Binding experiments

White pupae of w$^-$ flies were collected and kept at 25°C. The Eye-brain complexes were dissected at 40 hr APF in PBS and fixed in 2% paraformaldehyde (PFA) in PBS for 20 min. After the fixation, the samples were washed in PBS for 30 min with several exchange of the buffer and embedded in the diluted agar (5% low melting point agar in pure water). The temperature of the agar was 40–45 degree. Wait until it gets solid for 30 min-1 h. Samples were sliced using LinearSlicer (Dousaka EM) with 90–100 µm thickness (FREQUENCY 85–87 Hz, AMPLITUDE 1–1.2 mm, speed 1 mm/min). The swimming slices were picked up with brush and incubated with 100 µl PTP69D or LAR-PLAP fusion protein (kindly provided by Dr. Kai Zinn) supernatant for 1.5 hr at room temperature. Then, the samples were postfixed with 5% PFA in PBS without prewashing for 1 hr at room temperature. After washing the fixative with PBS (three changes) for 15 min and with 0.05% PBT (PBS+0.05% Triton X-100, three changes) for 15 min, the slices were incubated with anti-PLAP (1:500, abcam) and 24B10 (1:50, DSHB) in 0.05% PBT overnight at 4°C. After washing with 0.05% PBT (three changes) for 15 min, the brains are incubated for 3 hr at room temperature in secondary antibodies (1:400, Life technologies). The brains were washed again with 0.05% PBT (three changes) and with PBS (three changes) for 15 min, then soaked in Vectashield (Vector Laboratories) overnight at 4°C. The slices were mounted just before imaging.

## Assessment of photoreceptor axonal phenotypes

R7 photoreceptor axons were visualized with Rh4-mCD8GFP in confocal-stack images of whole-mount brains. (*Figures 1*, *3*, *4*, *6*, *7B* and *8*, *Figure 2—figure supplement 1*, *Figure 6—figure supplement 1*, *Figure 8—figure supplement 1*, *Figure 9—figure supplement 1*) the number of axons terminating each layer was counted manually using NIS-elements AR and the ratio of the axons terminating each layer was calculated. Although the terminals of R7 axons were clearly visible in lamina (see arrows in *Figure 1D*), a precise quantification was hampered by the overlapping nature of R7 photoreceptor axons inside the lamina. Therefore, we compared the number of R7 photoreceptor axon innervating medulla between the wild-type and the mutant flies. In the wild type, there are 21.7 Rh4-mCD8GFP positive R7 photoreceptor axons per 10 µm stack (466 µm in total thickness). We then calculate the expected number of R7 axons which should innervate medulla according to total thickness of scanned images. If the number of R7 axons innervating medulla in mutant flies is less than 90% of the expected number from the calculation above, we considered that the deficit is caused by the axons terminated in the lamina. Then we subtracted the number of R7 axons innervating medulla from the expected number and regarded the difference as the number of R7 axons terminated in the lamina. For example, the total number of the R7 axons innervating medulla in *LAR, ptp69D* double mutant was 102 (376µm in total thickness). The expected number of axons was calculated to be $21.7 \times 376/10 = 815.92$. As 102 is less than 90% of expected axons number ($815.92 \times 0.9 = 734.328$), we consider the difference ($815.92–102 = 713.92$) is the R7 axons in double mutant terminating in the lamina (87.5%). For R8 axons, the axons were visualized with Rh6- mCD8GFP and the ratio of the axons terminating each layer was calculated in the same manner as R7 (*Figures 5* and *7C*, *Figure 5—figure supplement 1*). In the wild type, there are 18.6 Rh6-mCD8GFP positive R8 photoreceptor axons per 10 µm stack (318 µm in total thickness).

## Acknowledgements

We thank Drs. Kaushiki Menon, Peter Hyung-Kook Lee and Kai Zinn for providing the reagents and technical instructions. We gratefully acknowledge Dr. Jessica Treisman, Dr. Tory Herman, Bloomington stock center, Kyoto DGRC, VDRC, Harvard stock center, and DSHB for providing fly or antibody stocks. This work was supported by Grant-in-Aid for JSPS Research Fellow (SH-S), JSPS KAKENHI Grant number 26440119 (SH-S) and 26291047 (TS), Grant-in-Aid for Scientific Research on Innovative

Areas from Ministry of Education, Culture, Sports, Science, and Technology of Japan 16H06457 'Dynamic Regulation of Brain Function by Scrap and Build System' (TS).

## Additional information

### Funding

| Funder | Grant reference number | Author |
| --- | --- | --- |
| Japan Society for the Promotion of Science | Grant-in-Aid for JSPS Research Fellow | Satoko Hakeda-Suzuki |
| Japan Society for the Promotion of Science | KAKENHI 26440119 | Satoko Hakeda-Suzuki |
| Japan Society for the Promotion of Science | KAKENHI 26291047 | Takashi Suzuki |
| Japan Society for the Promotion of Science | Grant-in-Aid for Scientific Research on Innovative Areas 16H06457 | Takashi Suzuki |
| Toray Industries | | Takashi Suzuki |

The funders had no role in study design, data collection and interpretation, or the decision to submit the work for publication.

### Author contributions

Satoko Hakeda-Suzuki, Conceptualization, Supervision, Funding acquisition, Investigation, Methodology, Writing—original draft, Project administration, Writing—review and editing; Hiroki Takechi, Investigation, Methodology; Hinata Kawamura, Investigation; Takashi Suzuki, Conceptualization, Supervision, Funding acquisition, Writing—original draft, Project administration, Writing—review and editing

### Author ORCIDs

Takashi Suzuki https://orcid.org/0000-0001-9093-2562

### Decision letter and Author response

Decision letter https://doi.org/10.7554/eLife.31812.029
Author response https://doi.org/10.7554/eLife.31812.030

## Additional files

### Supplementary files

• Supplementary file 1. Table showing the list of all full genotypes used.
DOI: https://doi.org/10.7554/eLife.31812.027

• Transparent reporting form
DOI: https://doi.org/10.7554/eLife.31812.028

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
