## [Decision Letter]

[Editors’ note: a previous version of this study was rejected after peer review, but the authors submitted for reconsideration. The first decision letter after peer review is shown below.]

Thank you for submitting your work entitled "Two receptor tyrosine phosphatases dictate the depth of axonal stabilizing layer in the visual system" for consideration by *eLife*. Your article has been reviewed by three peer reviewers, and the evaluation has been overseen by a Reviewing Editor and a Senior Editor. The following individuals involved in review of your submission have agreed to reveal their identity: Matthew Pecot (Reviewer #1) and Kai Zinn (Reviewer #3).

Our decision has been reached after consultation between the reviewers. Based on these discussions and the individual reviews below, we regret to inform you that your work will not be considered further for publication in *eLife*.

The reviewers were in general positive about your study showing that the two receptor protein tyrosine phosphatases (RPTPs), LAR and Ptp69D act in an overlapping manner to control specific targeting of R7 and R8 photoreceptors in the medulla neuropil of the fly eye. Your findings that active RPTP signaling is required for R7 and R8 axons to stabilize in the correct layer, and that these axons retract back to the lamina when both LAR and 69D are absent constitute an advance. The reviewers were impressed by your ability to produce photoreceptor axons that lack both genes and by the thorough rescue experiments you performed.

However, two of the three reviewers - and in the post-review discussion all three - agreed that experiments addressing cell autonomy are lacking and that this effort would firm up the story. They point out that LAR and Ptp69D are expressed broadly and in the RNAi experiments they are disrupted broadly. They call for genetic mosaic methods and cell-specific enhancers for manipulating gene function selectively in R7 or R8 neurons (both are not required).

In addition, you claim that the levels of LAR and Ptp69D expression and the degree of intracellular signaling play a key role in layer termination, based on RNAi knockdown or the overexpression of cDNA constructs containing or lacking specific signaling domains in rescue experiments. However, there is no assessment of the degree to which RNAi affect protein levels or whether rescue constructs are expressed equivalently. This issue could be addressed by the use of tagged constructs, and/or through immunostaining with fluorescence probes.

Another suggestion is to test the LAR mutant generated by Treisman that retains only FNIII repeats 7-9, unless it is no longer available, to see if rescue can ensue.

If additional experiments are predicted to take longer than 2 months, *eLife*'s policy is to reject the paper in its present form. If your efforts to demonstrate cell autonomous defects are successful, the paper could be again considered.

*Reviewer #1:*

Previous studies showed that *Drosophila* photoreceptor neurons (R1-R8) require the activities of two receptor protein tyrosine phosphatases (RPTPs), LAR and Ptp69D, for proper axon targeting within the lamina and medulla neuropils. Here the authors attempt to address how these molecules work together to regulate the layer-specific targeting of R7 and R8 photoreceptors in the medulla neuropil. They propose that these proteins act in an overlapping manner to control layer specificity. They claim that final layer termination depends on the combination of differential ligand recognition and the cumulative levels of intracellular signaling through a common pathway.

In general I think the lack of experiments addressing cell autonomy and levels of expression make it difficult to interpret the results, and leads me to feel the conclusions are overstated. This is particularly problematic because experiments similar to a number of those described here have been published in previous studies (e.g. LAR, Ptp69D single and double mutants, structure function experiments), and without performing more precise cell-specific manipulations it's not clear what the advance is.

1) LAR and Ptp69D are expressed broadly in the eye and brain, and required in all photoreceptors for targeting. Thus, a major concern is that in all experiments LAR and Ptp69D are disrupted broadly, either in the whole eye (RNAi) or in large patches in the eye and brain (FLICK), and rescue constructs are likewise expressed broadly in the whole eye, yet conclusions are made about the function of these molecules in R7 or R8 specifically. Non-autonomous effects cannot be ruled out, and it is essentially impossible to assess cellular requirements in these experiments. Genetic mosaic methods and cell-specific enhancers for manipulating gene function selectively in R7 or R8 neurons are well established and could be applied here to address these issues, although these experiments may be complicated genetically and could require more than two months to complete.

2) In addition, it is claimed that the levels of LAR and Ptp69D expression (Figure 3) and the degree of intracellular signaling (Figure 4) plays a key role in layer termination, based on RNAi knockdown or the overexpression of cDNA constructs containing or lacking specific signaling domains in rescue experiments. However, there was no assessment of the degree to which RNAi affected protein levels or whether rescue constructs were expressed equivalently. Thus, on top of the issue of cell autonomy, more controls are needed to make conclusions about the requirement for specific protein levels and signaling activities. LAR immunostaining could be assessed in RNAi experiments in the eye, and rescue constructs could potentially be tagged and visualized through immunostaining (quantify fluorescence). If tagged constructs do not exist then transgenic flies would have to be made making it unreasonable to complete such experiments in a two month window.

I feel strongly that the authors need to address the above issues before the paper would be suitable for publication in *eLife*.

*Reviewer #2:*

Layer-specific targeting of photoreceptor axons in fly visual system is a central model for understanding molecular mechanisms that guide axons to their targets in the brain. Work from a number of labs identified a pair of receptor tyrosine phosphatases, LAR and PTP69D, as playing important roles in stabilizing R7 photoreceptor axons in their target layer in the medulla. However, the mechanisms that determine how these phosphatases act, and might guide R cell axons past alternative target layers have remained unclear. Here Hakeda-Suzuki et al. make a significant contribution to our understanding by producing photoreceptor axons that lack both genes, and then performing an exhaustive series of rescue experiments using both genes. These studies reveal that phosphatase activity is required for axons to stabilize in the medulla, with different levels of activity causing targeting to different layers, patterns that correlate with the expression patterns of phosphatase ligands. Overall, this paper makes a valuable contribution to our understanding of the roles of receptor tyrosine phosphatases in layer-specific targeting, and I have no major concerns regarding the data.

*Reviewer #3:*

This is an interesting paper reporting a unique phenotype in which loss of both Lar and Ptp69D (69D) causes R7 axons to retract fully from the medulla. The paper does not define a mechanism that can account for the phenotypes, beyond what has been reported earlier by others. It was already known (or thought) that Lar and 69D form a complex in R7 axons, and that the signal from that complex can be transduced by either Ptp domain. Lar phosphatase activity is not required for normal R7 targeting because 69D phosphatase activity can substitute for Lar activity. Nevertheless, these are remarkable findings, given that they demonstrate that active RPTP signaling is required for R7 and R8 axons to stabilize in the correct layer, and that they retract back to the lamina when both Lar and 69D are absent.

1) In general, the authors fail to emphasize that most RPTP phenotypes are only observed in double or triple mutants. Thus, redundancy is the rule rather than the exception for these enzymes (summarized in Jeon et al. 2008). For example, embryonic longitudinal CNS axons project normally in 10D or 69D mutants, but cross the midline in 10D 69D double mutants (Sun et al. 2000). Tracheae develop normally in 10D or 4E mutants, but develop cysts and cannot inflate in 4E 10D double mutants (Jeon et al. 2009, 2012). ISNb motor axons project normally in 69D or 99A mutants, but are usually truncated in double mutants (Desai et al. 96). Finally, most relevant to the present study, some ISN motor axons usually reach their normal termination site in Lar mutants, but the bundle appears thinner. In Lar 69D double mutants, the ISN is truncated at its second branchpoint, and in Lar 69D 99A triple mutants it is truncated at the first branchpoint (Desai et al. 97). It thus gets shorter as RPTPs are removed, similarly to R axons (in light of these new studies, it would be interesting to go back and do live imaging in the embryo and see if the axons initially reach the normal termination site in mutants but then later retract). It doesn't make sense to say, as they do in the Discussion, that redundancy between Lar and 69D is a mechanism to ensure preservation of color vision. One might as well say that redundancy between 10D and 4E is a way to preserve respiration. The fly is insulated against some phenotypes by redundancy, but there are obviously many single mutant phenotypes that cause lethality.

2) They need to count R7s in the retina, not just show that some R7 axons remain in the double mutant. It is possible that many R7s die as a consequence of failing to synapse.

3) They should test the Lar mutant (generated by Treisman and colleagues) that retains only FNIII repeats 7-9, unless it is no longer available, to see if it rescues. Hofmeyer and Treisman showed that these repeats were sufficient to rescue the Lar single mutant phenotype.

4) The Lar and 69D-AP staining is not very informative. It probably represents the cumulative expression of multiple ligands. While there is staining in M6 for Lar-AP, it is not the most prominent staining, and 69D-AP weakly stains all layers. It is possible that the staining of M6 by 69D-AP is slightly lower than that of other layers, but this is not very convincing. Still, it seems acceptable to include these data in a supplement. At least they did the experiment.

5) Figure 4: It appears that the DM phenotype can be substantially rescued by LAR 2XCS, implying that neither Lar or 69D phosphatase activity is absolutely necessary (if both mutants are nulls). However, binding of something to Lar D2 appears to be required, as has been observed earlier by other groups. Oddly, however, for 69D to rescue the DM phenotype phosphatase activity of one domain is required. However, this assumes that 69D D2 does have enzymatic activity and that the DA mutation is equivalent to a CS mutation; it is also possible that the DA1DA2 double mutant acts as a substrate trap and makes the phenotype worse via some kind of dominant negative (DN) effect. Expressing the DA1DA2 construct in wild-type embryonic neurons produces axonal phenotypes.

6) Figure 5 axons usually retract only to M1, not to lamina, in DM. This is interesting. LarΔC acts as a DN in DM, doubling the fraction that retract to lamina. Why? This figure also shows that the distinction between Lar stabilizing axons at M6 and 69D stabilizing them at M3 is not as clean as they wish to argue. LarFL overexpression in the DM causes 50% of R8s to go to M6, and LarD1CS causes about 20% to go to M6. For 69D, full-length 69D causes 20% to go to M6, indicating that it has the same effect as Lar D1CS.

7) Figure 6: It is odd that Ptp69D-deltaintra seems to be able to block most Lar activity (since it generates a phenotype almost as strong as the DM when expressed in a 69D mutant), and can completely block 69D activity (since it generates a DM phenotype when expressed in a Lar mutant), yet it generates no phenotype on its own when it is expressed in a wild-type background. Perhaps in this case there is residual Lar activity that can maintain most axons at M6. Still you would think that you would get a 69D-like phenotype.

[Editors’ note: what now follows is the decision letter after the authors submitted for further consideration.]

Thank you for resubmitting your work entitled "Two receptor tyrosine phosphatases dictate the depth of axonal stabilizing layer in the visual system" for further consideration at *eLife*. Your revised article has been favorably evaluated by K VijayRaghavan (Senior Editor), a Reviewing Editor, and three reviewers.

The manuscript has been improved but there are some remaining issues that need to be addressed before acceptance, as outlined below:

1) On the issue of cell autonomy: In the experiment in which you use a combination of MARCM and RNAi, PTP69D and Lar in R7 were knocked out. MARCM was used to generate single R7 neurons (and R1 andR6 neurons) that are null for PTP69D in an otherwise heterozygous background. A Lar null mutation was included in the background and whole eye (GMR) Lar RNAi was used in addition to knockdown of Lar function.

a) Please clarify what the control genotype is.

b) Because R1 and R6 clones are also generated, and these neurons terminate axons in the lamina similar to the double knockout R7 neurons, it was not possible to directly identify R7 axons mis-targeting to the lamina. Instead, the quantification is based on the number of axons innervating the medulla in control versus double knockout and the difference between them is assumed to represent R7 axons terminating in the lamina. This may be correct, but it is also possible that there was differential Flping between the genotypes (i.e. different numbers of clones generated in wild type and mutant backgrounds). A simple way to resolve this without changing the genetics would be to address the number of clones in wt and DKO that innervate the medulla in early pupal development, before the mutant neurons retract to the lamina. At this stage the number of clones in the medulla should be similar, which would be consistent with your hypothesis.

2) Your experiments on cell-specific rescue experiments using the GAL4/UAS system and FL Lar constructs are compelling but references for the specificity of the drivers and evidence of cell-specific expression should be provided.

3) As you have not addressed the issue of directly assessing protein levels for their expressed constructs, this caveat should be acknowledged in your experiments.

4) Explain what the n's mean in Figure 7. Are these number of stacks analyzed, and or the number of total axons? If there were only 4 or 5 clones total, that would not be sufficient. But those numbers could not explain the statistics, as you can't get 5% penetrance with 4 axons. If the n's refer to the number of stacks analyzed, and the micron values to the depth of those stacks, why are there no numbers associated with the DM animals in D-F and G-I? The only numbers are for the rescue, but the unrescued DMs appear to be different animals than those analyzed in A-C. They have to be different for G-I, as here one is viewing R8 rather than R7. This should be clarified.

---

## [Author Response]

[Editors’ note: the author responses to the first round of peer review follow.]

However, two of the three reviewers - and in the post-review discussion all three - agreed that experiments addressing cell autonomy are lacking and that this effort would firm up the story. They point out that LAR and Ptp69D are expressed broadly and in the RNAi experiments they are disrupted broadly. They call for genetic mosaic methods and cell-specific enhancers for manipulating gene function selectively in R7 or R8 neurons (both are not required).

See response to reviewer #1, comment #1.

In addition, you claim that the levels of LAR and Ptp69D expression and the degree of intracellular signaling play a key role in layer termination, based on RNAi knockdown or the overexpression of cDNA constructs containing or lacking specific signaling domains in rescue experiments. However, there is no assessment of the degree to which RNAi affect protein levels or whether rescue constructs are expressed equivalently. This issue could be addressed by the use of tagged constructs, and/or through immunostaining with fluorescence probes.

See response to reviewer #1, comment #2.

Another suggestion is to test the LAR mutant generated by Treisman that retains only FNIII repeats 7-9, unless it is no longer available, to see if rescue can ensue.

See response to reviewer #3, comment #3.

Reviewer #1:[…] 1) LAR and Ptp69D are expressed broadly in the eye and brain, and required in all photoreceptors for targeting. Thus, a major concern is that in all experiments LAR and Ptp69D are disrupted broadly, either in the whole eye (RNAi) or in large patches in the eye and brain (FLICK), and rescue constructs are likewise expressed broadly in the whole eye, yet conclusions are made about the function of these molecules in R7 or R8 specifically. Non-autonomous effects cannot be ruled out, and it is essentially impossible to assess cellular requirements in these experiments. Genetic mosaic methods and cell-specific enhancers for manipulating gene function selectively in R7 or R8 neurons are well established and could be applied here to address these issues, although these experiments may be complicated genetically and could require more than two months to complete.

First of all, we thank the reviewers for the comment. According to this valuable suggestion, we have addressed this issue with additional analysis. We have added 2 ways to demonstrate the cell autonomy of the RPTPs overlapping function. One for R7-specific loss-of- function, and another for R7-specific rescue experiment.

The detailed genotype is:

Double mutant R7 single cell clone: GMR-FLP, UAS-mCD8::GFP/+; GMR-Gal4, LAR[2127]/ UAS-LARRNAi; Ptp69D[D1689] FRT80B / GMR-mCD8-KO-myc, tub-Gal80, FRT80B,

R7-specific rescue: yw, eyFLP2 (c-lacZ)/PM181-Gal4; LAR[2127] /<LAR<, Rh4mCD8::GFP; Ptp69D[D1689] /<Ptp69D< UAS-LARFL

R8-specific rescue: yw, eyFLP2 (c-lacZ)/2-80Gal4; LAR[2127] /<LAR<, Rh6- mCD8::GFP; Ptp69D[D1689] /<Ptp69D< UAS-LARFL

In the former experiment, 70% of double mutant R7 showed retraction to the lamina, whereas only 25% of them correctly targeted to the M6 layer. Less frequent ratio of retraction compared to the eyFLP clone is probably due to the perdurance of the RPTP protein where the clone was induced at the last step of differentiation, compared to the eyFLP (all photoreceptors) mutant FLICK clone. Nonetheless, significant defect that was observed in single R7 clone strongly suggests RPTPs’ cell-autonomous function.

In the latter experiment, we tried to rescue eyFLP double FLICK mutant by R7-specific LAR expression. There, we observed around 80% innervated the medulla and 50% to the M6 layer. This also showed lower success rate compared to the GMR rescue. We reasoned that PM181-Gal4 might be insufficient to rescue 100%, either too weak or too early to stop expressing UAS-LAR, therefore we concluded that overall 80% is a sufficient number to claim the cell-autonomous function in R7 axons. We also performed an R8-specific rescue, which resulted in 100% rescue, therefore concluded that RPTP function in R8 in cell autonomously. In R8, probably the expression of the 2-80-Gal4 was consistent to cover the critical period to rescue. We have summarized these data in a figure and newly named as the Figure 7.

2) In addition, it is claimed that the levels of LAR and Ptp69D expression (Figure 3) and the degree of intracellular signaling (Figure 4) plays a key role in layer termination, based on RNAi knockdown or the overexpression of cDNA constructs containing or lacking specific signaling domains in rescue experiments. However, there was no assessment of the degree to which RNAi affected protein levels or whether rescue constructs were expressed equivalently. Thus, on top of the issue of cell autonomy, more controls are needed to make conclusions about the requirement for specific protein levels and signaling activities. LAR immunostaining could be assessed in RNAi experiments in the eye, and rescue constructs could potentially be tagged and visualized through immunostaining (quantify fluorescence). If tagged constructs do not exist then transgenic flies would have to be made making it unreasonable to complete such experiments in a two month window.

We have addressed the issue of assessing the expression level of the RNAi and the rescuing transgenes. We have added 2 supplementary data using antibody staining to show that the RNAi knock down have successfully down-regulated the protein level in desired manner, and that transgene expression was monitored to show that they are all expressed in considerably equivalent and sufficient manner, and the rescuing ability is not correlated with the expression level, but rather with the protein domains and the mutations. Altogether, we have now successfully demonstrated that the levels of LAR and Ptp69D expression and the degree of intracellular signaling play a key role in layer termination.

These are shown in Figure 3—figure supplement 1 and Figure 4—figure supplement 1.

Reviewer #3:[…] 1) In general, the authors fail to emphasize that most RPTP phenotypes are only observed in double or triple mutants. Thus, redundancy is the rule rather than the exception for these enzymes (summarized in Jeon et al. 2008). For example, embryonic longitudinal CNS axons project normally in 10D or 69D mutants, but cross the midline in 10D 69D double mutants (Sun et al. 2000). Tracheae develop normally in 10D or 4E mutants, but develop cysts and cannot inflate in 4E 10D double mutants (Jeon et al. 2009, 2012). ISNb motor axons project normally in 69D or 99A mutants, but are usually truncated in double mutants (Desai et al. 96). Finally, most relevant to the present study, some ISN motor axons usually reach their normal termination site in Lar mutants, but the bundle appears thinner. In Lar 69D double mutants, the ISN is truncated at its second branchpoint, and in Lar 69D 99A triple mutants it is truncated at the first branchpoint (Desai et al. 97). It thus gets shorter as RPTPs are removed, similarly to R axons (in light of these new studies, it would be interesting to go back and do live imaging in the embryo and see if the axons initially reach the normal termination site in mutants but then later retract). It doesn't make sense to say, as they do in the Discussion, that redundancy between Lar and 69D is a mechanism to ensure preservation of color vision. One might as well say that redundancy between 10D and 4E is a way to preserve respiration. The fly is insulated against some phenotypes by redundancy, but there are obviously many single mutant phenotypes that cause lethality.

We thank the reviewer, and have stated the redundancy of phosphatases in the Introduction more precisely. We also removed the last paragraph in the Discussion (redundancy between Lar and 69D is a mechanism to ensure preservation of color vision) as this was pointed out by the reviewer 1 as well.

2) They need to count R7s in the retina, not just show that some R7 axons remain in the double mutant. It is possible that many R7s die as a consequence of failing to synapse.

We confirmed the survival of the R7 cells and axon extending from those cells by agarose section staining.

3) They should test the Lar mutant (generated by Treisman and colleagues) that retains only FNIII repeats 7-9, unless it is no longer available, to see if it rescues. Hofmeyer and Treisman showed that these repeats were sufficient to rescue the Lar single mutant phenotype.

We have done the rescue experiment using the LAR mutant generated by J. Treisman that retains only FNIII repeats 7-9, and successfully rescue the double mutant phenotype. We included this data in Figure 4.

4) The Lar and 69D-AP staining is not very informative. It probably represents the cumulative expression of multiple ligands. While there is staining in M6 for Lar-AP, it is not the most prominent staining, and 69D-AP weakly stains all layers. It is possible that the staining of M6 by 69D-AP is slightly lower than that of other layers, but this is not very convincing. Still, it seems acceptable to include these data in a supplement. At least they did the experiment.

We keep including these data in the supplemental data.

5) Figure 4: It appears that the DM phenotype can be substantially rescued by LAR 2XCS, implying that neither Lar or 69D phosphatase activity is absolutely necessary (if both mutants are nulls). However, binding of something to Lar D2 appears to be required, as has been observed earlier by other groups. Oddly, however, for 69D to rescue the DM phenotype phosphatase activity of one domain is required. However, this assumes that 69D D2 does have enzymatic activity and that the DA mutation is equivalent to a CS mutation; it is also possible that the DA1DA2 double mutant acts as a substrate trap and makes the phenotype worse via some kind of dominant negative (DN) effect. Expressing the DA1DA2 construct in wild-type embryonic neurons produces axonal phenotypes.

We have tested whether expressing the DA1DA2 construct in wild-type photoreceptor neurons produces axonal phenotypes, but the result was negative (normal), suggesting that DA1DA2 does not act as a dominant negative in the photoreceptors. But we fully agree with the reviewer that the observation we made here, and that of J. Treisman’s study (PNAS 2010), contains some mysteries that we still cannot explain very well. Differential requirement of phosphatase domain between LAR and Ptp69D in a context dependent manner is one of those that we should further clarify in the future.

6) Figure 5 axons usually retract only to M1, not to lamina, in DM. This is interesting. LarΔC acts as a DN in DM, doubling the fraction that retract to lamina. Why? This figure also shows that the distinction between Lar stabilizing axons at M6 and 69D stabilizing them at M3 is not as clean as they wish to argue. LarFL overexpression in the DM causes 50% of R8s to go to M6, and LarD1CS causes about 20% to go to M6. For 69D, full-length 69D causes 20% to go to M6, indicating that it has the same effect as Lar D1CS.

We agree with the reviewer that those points made by the reviewer are fully justified, and we cannot make a strong argument, but can only speculate here.

LAR∆C acts DN in R8: It is possible that, only in R8, there is the third RPTP that can bind to LAR ectodomain, and is sharing the downstream signaling with LAR. However, this is just a pure speculation.

Ptp69D has also M6 targeting effect: We fully agree with the reviewer. Our hypothesis was to satisfy the largest number of observations we made, but we admit that we fail to do so for a small number of observations. Future studies and a theory hopefully explain everything.

7) Figure 6: It is odd that Ptp69D-deltaintra seems to be able to block most Lar activity (since it generates a phenotype almost as strong as the DM when expressed in a 69D mutant), and can completely block 69D activity (since it generates a DM phenotype when expressed in a Lar mutant), yet it generates no phenotype on its own when it is expressed in a wild-type background. Perhaps in this case there is residual Lar activity that can maintain most axons at M6. Still you would think that you would get a 69D-like phenotype.

We fully agree with the reviewer that it seems quite odd. The explanation we came up is as follows: As a prerequisite, we set the following conditions.

1) LAR-69D complex contains more than three proteins, possibly tetramer (?).

2) In the wild type situation, when LAR ligand binds to LAR-69D complex, only the cytoplasmic domain of 69D gets activated.

3) If the complex contains both 69D(FL) and LAR(FL), 69D∆Intra cannot interfere with their signaling. Therefore, overexpression of 69D∆Intra in wild type background does not create any defects.

4) If there is only LAR (in 69D mutant), 69D_∆_Intra binds to LAR(FL) and activates only the cytoplasmic domain of 69D, which does not exist here in this situation. As a result, 69D∆Intra suppress the activity of LAR cytoplasmic domain and therefore act as a DN.

5) If there is only 69D (in LAR mutant), 69D_∆_Intra sequestrates the 69D ligand, and therefore act as a DN.

In these conditions, there can be a difference when ptp69D-∆intra was expressed in wild type background (both LAR and 69D are available) or in Ptp69D mutant (only LAR is available).Since this is a pure speculation, we did not describe this in the Results or in Discussion.

[Editors' note: the author responses to the re-review follow.]

1) On the issue of cell autonomy: In the experiment in which you use a combination of MARCM and RNAi, PTP69D and Lar in R7 were knocked out. MARCM was used to generate single R7 neurons (and R1 andR6 neurons) that are null for PTP69D in an otherwise heterozygous background. A Lar null mutation was included in the background and whole eye (GMR) Lar RNAi was used in addition to knockdown of Lar function.a) Please clarify what the control genotype is.

The exact genotype is/was shown in Supplementary file 1. The genotype of the control is:

GMR-FLP UAS-mCD8GFP/+: GMR-Gal4/+; FRT80/ tub-Gal80 GMR-mCD8KOmyc FRT80

Additionally, we have inserted the exact genotype in the legend of Figure 7, so that the readers can understand it without going to the supplementary figures.

b) Because R1 and R6 clones are also generated, and these neurons terminate axons in the lamina similar to the double knockout R7 neurons, it was not possible to directly identify R7 axons mis-targeting to the lamina. Instead, the quantification is based on the number of axons innervating the medulla in control versus double knockout and the difference between them is assumed to represent R7 axons terminating in the lamina. This may be correct, but it is also possible that there was differential Flping between the genotypes (i.e. different numbers of clones generated in wild type and mutant backgrounds). A simple way to resolve this without changing the genetics would be to address the number of clones in wt and DKO that innervate the medulla in early pupal development, before the mutant neurons retract to the lamina. At this stage the number of clones in the medulla should be similar, which would be consistent with your hypothesis.

When we considered about how to quantify the Flping efficiency of GMR-FLP in the mutant and the control, we realized that we could utilize our existing data. As we applied the cMARCM system (Tomasi et al., 2008), the WT axons were labeled with GMR-mCD8-Kusabira Orange (GMR-mCD8KO). The lack of KO signal indicates that the axon is a GFP positive clone. We thereby quantified the gaps in M6 terminating axons of GMR-mCD8KO (arrowheads in Figure 7) and confirmed that the number of the double mutant R7 clones was comparable with the control (control: 22.3/100μm, DM: 27.4/100μm).

We have added a sentence in the Results section to explain this result as follows:

“As we applied the cMARCM system (Tomasi et al., 2008), the WT axons were labeled with GMR-mCD8-Kusabira Orange (GMR-mCD8KO). […] We thereby quantified the gaps in M6 terminating axons of GMR-mCD8KO (arrowheads in Figure 7) and confirmed that the number of the double mutant R7 clones was comparable with the control (control: 22.3/100μm, DM: 27.4/100μm).”

2) Your experiments on cell-specific rescue experiments using the GAL4/UAS system and FL Lar constructs are compelling but references for the specificity of the drivers and evidence of cell-specific expression should be provided.

We have made an additional supplementary figure (Figure 7—figure supplement 1) to show the specific expression of the drivers we used for the R7 and R8-specific rescue. 2-80-Gal4 is a gift from Tory Herman, but it is not published before, therefore we added the explanation in the Materials and methods section as follows. “2-80-Gal4 is a Gal4 enhancer trap line on the X chromosome kindly provided by Tory Herman. R8-specific expression persists until mid-pupal stage.” PM181-Gal4 is already published in a number of papers (Lee et al., 2001, Maurel-Zaffran et al., 2001).

3) As you have not addressed the issue of directly assessing protein levels for their expressed constructs, this caveat should be acknowledged in your experiments.

Actually, we have addressed this issue by antibody staining, and 2 additional supplementary figures (Figure 3—figure supplement 1 and Figure 4—figure supplement 1) were added before the resubmission stage. We could not confidently understand what the issue is, but we assumed that the problem is that we have not explained the additional experiments we did in the Results section. Therefore, we have added the following sentence in the Results section:

“We have assessed the expression level of the rescuing transgenes using antibody staining against larval eye discs. […] Altogether, we have now successfully demonstrated that the degree of intracellular signaling play a key role in layer termination.”

4) Explain what the n's mean in Figure 7. Are these number of stacks analyzed, and or the number of total axons? If there were only 4 or 5 clones total, that would not be sufficient. But those numbers could not explain the statistics, as you can't get 5% penetrance with 4 axons. If the n's refer to the number of stacks analyzed, and the micron values to the depth of those stacks, why are there no numbers associated with the DM animals in D-F and G-I? The only numbers are for the rescue, but the unrescued DMs appear to be different animals than those analyzed in A-C. They have to be different for G-I, as here one is viewing R8 rather than R7. This should be clarified.

“n” indicates the number of independent hemispheres of the brain analyzed and the depth of the total stacks are also indicated as μm. As it seems not clear enough, we also indicate the total axon number in Figure 7. As it was pointed out by the reviewer, we have inserted the number of the hemisphere “n”, and the stack depth as microns, for the DM as well in the Figure 7.